# A systematic CRISPR screen reveals an IL-20/IL20RA-mediated immune crosstalk to prevent the ovarian cancer metastasis

Jia Li[1†], Xuan Qin[1†], Jie Shi[1], Xiaoshuang Wang[1], Tong Li[2], Mengyao Xu[1], Xiaosu Chen[1], Yujia Zhao[1], Jiahao Han[1], Yongjun Piao[1], Wenwen Zhang[3], Pengpeng Qu[4], Longlong Wang[1], Rong Xiang[1]*, Yi Shi[1]*

[1]The School of Medicine, Nankai University, Tianjin, China; [2]Department of Lung Cancer Surgery, Tianjin Medical University General Hospital, Tianjin, China; [3]Research Institute of Obstetrics and Gynecology, Tianjin Central Hospital of Obstetrics and Gynecology, Tianjin, China; [4]Department of Gynecological Oncology, Tianjin Central Hospital of Obstetrics and Gynecology, Tianjin, China

**Abstract** Transcoelomic spread of cancer cells across the peritoneal cavity occurs in most initially diagnosed ovarian cancer (OC) patients and accounts for most cancer-related death. However, how OC cells interact with peritoneal stromal cells to evade the immune surveillance remains largely unexplored. Here, through an in vivo genome-wide CRISPR/Cas9 screen, we identified IL20RA, which decreased dramatically in OC patients during peritoneal metastasis, as a key factor preventing the transcoelomic metastasis of OC. Reconstitution of IL20RA in highly metastatic OC cells greatly suppresses the transcoelomic metastasis. OC cells, when disseminate into the peritoneal cavity, greatly induce peritoneum mesothelial cells to express IL-20 and IL-24, which in turn activate the IL20RA downstream signaling in OC cells to produce mature IL-18, eventually resulting in the polarization of macrophages into the M1-like subtype to clear the cancer cells. Thus, we show an IL-20/IL20RA-mediated crosstalk between OC and mesothelial cells that supports a metastasis-repressing immune microenvironment.

*For correspondence:
rxiang@nankai.edu.cn (RX);
yishi@nankai.edu.cn (YS)

[†]These authors contributed equally to this work

**Competing interests:** The authors declare that no competing interests exist.

## Introduction

Ovarian cancer (OC) has the highest mortality among all gynecological malignancies that seriously threatens women's health worldwide (*Siegel et al., 2020*). Among OC, epithelial ovarian cancer (EOC) is the most common type accounting for 90% of all cases (*Farley et al., 2008*). EOC is classified by tumor cell histology as serous (52%), endometrioid (10%), mucinous (6%), clear cell (6%), and other rare subtypes. Due to the lack of clear symptoms at the early stage, the spread of cancer cells across the peritoneal cavity occurs in most patients at the initial diagnosis, with approximately 70% of OC patients already at metastatic stage (stages III and IV) (*Vaughan et al., 2011*). For OC patients with metastasized cancer cells, current therapies including chemotherapy and targeted therapies can only achieve very limited clinical outcome, with the 5-year survival rate of 20–40% for OC patients at advanced stages (*Torre et al., 2018*).

Although OC cells can metastasize to other distant organs through blood vessels and lymphatic vessels, the transcoelomic metastasis of OC occurs most commonly, which includes multiple processes, such as detachment of tumor cells from primary sites, immune evasion of disseminated tumor cells in the peritoneal cavity, and colonization of tumor cells on the omentum and peritoneum (*Tan et al., 2006*). Disseminated OC cells have to survive in a peritoneal environment constituted by lymphocytes, macrophages, natural killer (NK) cells, fibroblasts, mesothelial cells, as well as cytokines and chemokines secreted by these cells for a successful transcoelomic metastasis (*Ahmed and*

*Stenvers, 2013*; *Worzfeld et al., 2017*). Although the role of the immune microenvironment in peritoneal cavity is indisputable in the metastasis of OC, the molecular network regulating the crosstalk between the disseminated tumors cells and immune microenvironment in peritoneal cavity is still just a tip of the iceberg. Therefore, it is imperative to understand the specific immune microenvironment in the peritoneal cavity for developing more efficient immunotherapeutic strategies against metastatic OC.

To systematically identify key genes involved in the regulation of the peritoneal metastasis of OC, we performed genome-wide gene knockout screening in an orthotopic mouse model of OC by using CRISPR/Cas9 knockout library, which has been successfully utilized to discover novel genes in the occurrence and development of diseases, especially in cancers (*Huang et al., 2019*; *Ng et al., 2020*; *Shalem et al., 2014*). We identified interleukin 20 receptor subunit alpha (IL20RA) as a potent suppressor of the transcoelomic metastasis of OC. IL20RA is mainly expressed in epithelial cells and forms the functional receptor, when heterodimerized with interleukin 20 receptor subunit beta (IL20RB), to bind immune cell-produced IL-20 subfamily of cytokines IL-19, IL-20, and IL-24, which have essential roles in regulating epithelial innate immunity and tissue repair (*Rutz et al., 2014*). IL20RA and IL20RB are also detected in tumors of epithelial origin including breast cancer, non-small-cell lung cancer, and bladder cancer, while IL-20 subfamily of cytokines has been reported to have either tumor-promoting or tumor-suppressing roles depending on tumor types and the local immune environment (*Gopalan et al., 2007*; *Lee et al., 2013*; *Pestka et al., 2004*; *Rutz et al., 2014*; *Whitaker et al., 2012*). In the present study, we discovered a novel IL-20/IL20RA-mediated crosstalk between disseminated OC cells and peritoneum mesothelial cells that eventually promotes the generation of M1-like inflammatory macrophages to prevent peritoneal dissemination of OC cells.

## Results

### *IL20RA* is an anti-transcoelomic metastasis gene in OC identified by a genome-scale knockout screening in vivo

To screen key genes regulating transcoelomic metastasis of OC, we utilized human epithelial OC cell SK-OV-3 with relatively low metastatic capacity to set up the orthotopic transplant tumor model in NOD-SCID mice. Prior to the inoculation of these cells into the mouse ovaries, SK-OV-3 cells were transduced with the human CRISPR knockout library (GeCKO v2.0) made by Feng Zhang et al., which contains sgRNAs specifically against 19,050 protein-encoding genes and 1864 miRNA genes and 1000 non-targeting control sgRNAs (*Shalem et al., 2014*). Peritoneal metastasized SK-OV-3 cells in the intraperitoneal cavity were isolated and expanded in vitro for next runs of orthotopic transplant (*Figure 1A*). After three runs of in vivo screening, highly metastatic SK-OV-3 cells were subjected to high-throughput sgRNA library sequencing to reveal the sgRNA representations. The RNAi Gene Enrichment Ranking (RIGER) p-value analysis was used to identify significantly enriched sgRNAs in metastasized (sgRNA$^{Met}$) or primary (sgRNA$^{Pri}$) OC xenografts. By using the criteria of the number of enriched sgRNAs targeting each gene $\geq 3$, p-value<0.05 and the normalized enrichment score (NES) $<-1.2$, we got two high-ranked genes, namely *IL20RA* and *TEX14* (*Figure 1B*). Given all the six sgRNAs targeting IL20RA in GeCKO library were enriched with high NES, we chose IL20RA for further investigation on its function in the transcoelomic metastasis of OC.

To confirm the role of IL20RA in OC metastasis, we knocked down IL20RA in SK-OV-3 cells, which had relatively high level of IL20RA (*Figure 1C, Figure 1—figure supplement 1A, B*), and reconstituted IL20RA in highly metastatic murine epithelial OC cell ID8, which was isolated from peritoneal ascites of ID8-injected C57BL/6 mice as described by Ward and colleagues (*Ward et al., 2013*) with high-grade serous ovarian carcinoma (HGSOC)-like characteristics (*Diaz Osterman et al., 2019*) and had very low level of endogenous IL20RA (*Figure 1—figure supplement 1A, Figure 1D*). In the SK-OV-3 orthotopic mouse OC model, silencing IL20RA greatly promoted the transcoelomic metastasis of OC and the formation of ascites (*Figure 1E, F*), while in ID8 syngeneic mouse OC model, reconstitution of IL20RA in ID8 cells dramatically reduced the volumes of ascites and the numbers of metastatic nodules in diaphragm, peritoneum, and mesentery (*Figure 1G, H*).

To further investigate whether IL20RA suppresses transcoelomic metastasis of OC at the later stage when OC cells have disseminated into the peritoneal cavity, we directly injected ID8 cells into

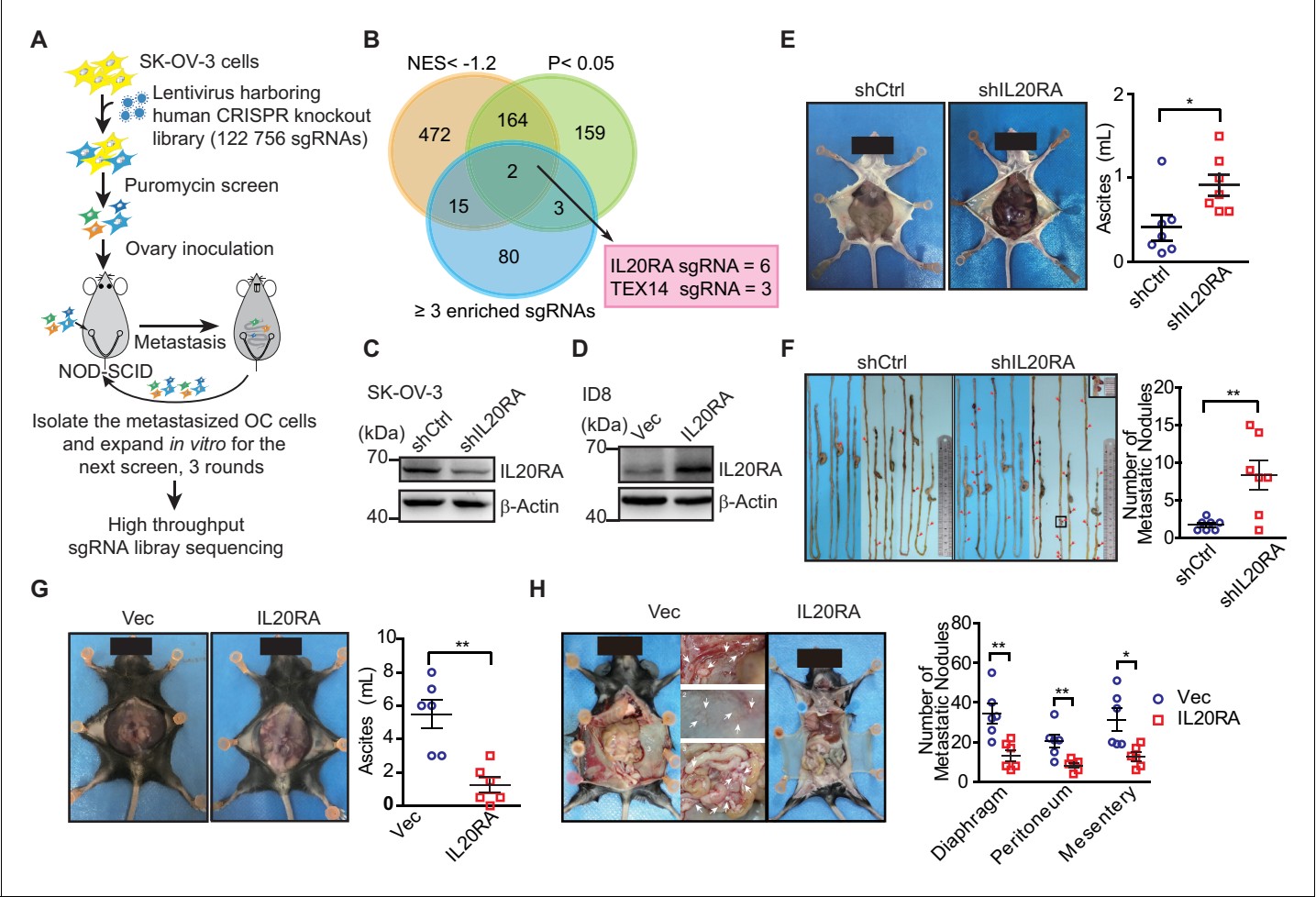

**Figure 1.** High-throughput CRISPR screen identified IL20RA as a suppressor of the transcoelomic metastasis of ovarian cancer (OC). (**A**) Schematics of experiment design to screen metastasis-related genes using CRISPR/Cas9 library in OC orthotopic murine model. (**B**) Venn diagram comparing the hits met the indicated enrichment criteria. (**C, D**) Western blot analysis of IL20RA in control shRNA (shCtrl)- or IL20RA shRNA (shIL20RA)-transfected SK-OV-3 cells (**C**) and in IL20RA- or empty vector-transfected ID8 cells (**D**). (**E, F**) Representative images of NOD-SCID mice with shCtrl- or shIL20RA-transfected SK-OV-3 cells orthotopically transplanted in the ovaries at 40 days post-inoculation (**E**, left panel). The ascites volumes (**E**, right panel) and the numbers of metastatic nodules on the surfaces of intestines (**F**) were quantified (n = 7, data are shown as means ± SEM, *p<0.05, **p<0.01, by unpaired two-sided Student's t-test). (**G, H**) Representative images of C57BL/6 mice at 60 days after orthotopically inoculated with IL20RA-reconstituted or control ID8 cells in ovaries (**G**, left panel). The ascites volumes (**G**, right panel) and the numbers of metastatic nodules on the surfaces lining the peritoneal cavities (**H**) were quantified (n = 6, data are shown as means ± SEM, *p<0.05, **p<0.01 by unpaired two-sided Student's t-test).

The online version of this article includes the following source data and figure supplement(s) for figure 1:

**Source data 1.** An Excel sheet with numerical quantification data.

**Figure supplement 1.** IL20RA suppresses the transcoelomic metastasis of ovarian cancer (OC) in syngeneic intraperitoneal OC mouse model.

the peritoneal cavity of C57BL/6 mice. We were still able to observe that IL20RA-reconstituted ID8 cells formed much less metastases on the diaphragm, peritoneum, and mesentery and resulted in much less ascites as well (*Figure 1—figure supplement 1C, D*). Together, these data suggest that IL20RA is a potent suppressor gene for the transcoelomic metastasis of OC.

## The IL20RA expression is dramatically decreased in metastases of OC patients and positively correlates with the clinical outcome

To further get the clinical evidences on the relevance of IL20RA in OC metastasis, we collected primary OC tissues and paired cancer cells isolated from ascites and metastatic nodules in peritoneal cavity from 20 serous OC patients. Analysis of IL20RA protein by western blot shows a dramatic decrease of IL20RA in the transcoelomic spread cancer samples when compared with those from the

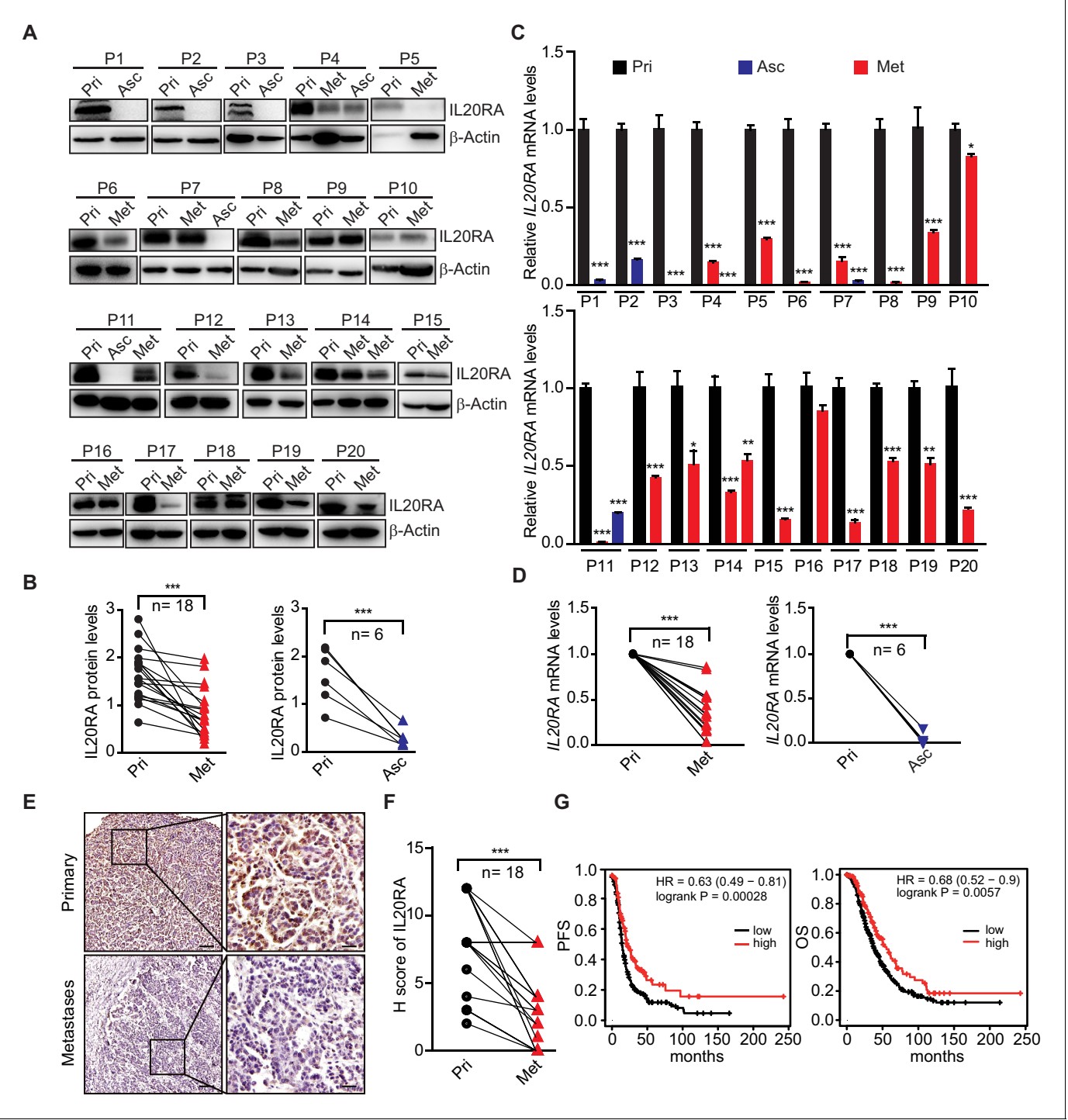

**Figure 2.** Dramatically decreased IL20RA in human ovarian cancer (OC) peritoneal metastases and its correlation with the clinical outcome. (A, B) Western blot analysis of IL20RA in primary OC tissues (Pri) and paired metastatic cancer cells in ascites (Asc) and metastatic nodules on the surfaces of the abdominal organs (Met) (A). Quantification results are plotted in (B) (n = 20, ***p<0.001, by unpaired two-sided Student's t-test). (C, D) qRT-PCR analysis of *IL20RA* in human primary OC tissue (Pri) and paired peritoneal metastases (Asc and Met) (C, data are plotted as means ± SEM from three independent measurements, *p<0.05, **p<0.01, ***p<0.001, by unpaired two-sided Student's t-test). The comparison of the *IL20RA* levels in these two groups is analyzed in (D) (**p<0.01, ***p<0.001, by unpaired two-sided Student's t-test). (E, F) Representative images of immunohistochemical analysis of IL20RA in in human primary OC tissue and paired peritoneal metastases (E) and quantification by H-score (F, n = 18, ***p<0.001, by paired two-sided Student's t-test). Scale bar: 100 μm (left panel in E); 20 μm (right panel in E). (G) Kaplan–Meier survival plot to show the progression-free survival and overall survival of serous OC patients with different IL20RA expression.

*Figure 2 continued on next page*

*Figure 2 continued*

The online version of this article includes the following source data and figure supplement(s) for figure 2:

**Source data 1.** An Excel sheet with numerical quantification data.
**Figure supplement 1.** IL20RA expression is positively correlated with the clinical outcome of patients in many types of cancers.
**Figure supplement 2.** IL20RB expression in human ovarian cancer (OC) tissues and its correlation with clinical outcome of OC patients.

primary sites (*Figure 2A, B*), which is confirmed by the quantitative reverse-transcriptase-PCR (qRT-PCR) analysis of *IL20RA* mRNA (*Figure 2C, D*). Immunohistochemical (IHC) analysis also shows the much lower level of IL20RA in metastatic cancer cells than that in the cancer cells at the primary sites (*Figure 2E, F*). To further investigate the relevance of IL20RA with OC metastasis, we analyzed the correlation between the IL20RA level and the survival of serous OC patients given that metastasis accounts for over 90% of cancer death. High level of IL20RA significantly correlates with the better overall survival (OS) and progression-free survival (PFS) of serous OC patients (*Figure 2G*), supporting that IL20RA functions as a suppressor of OC metastasis. Besides, IL20RA expression is also positively correlated with the clinical outcome of patients in patients of bladder carcinoma, uterine corpus endometrial carcinoma, rectum adenocarcinoma, and gastric cancer (*Figure 2—figure supplement 1*).

By contrast, as a heterodimerization partner for IL20RA, IL20RB expression shows no significant difference between primary and metastatic OC specimen (*Figure 2—figure supplement 2A, B*) and negatively correlates with the OS and PFS of OC patients (*Figure 2—figure supplement 2C*), suggesting that IL20RA is the key subunit of IL20RA/IL20RB receptor complex that is regulated during the transcoelomic metastasis of OC.

## Characterization of IL20RA-mediated modulation on the peritoneal immune microenvironment during the transcoelomic metastasis of OC

To get insights into the mechanisms of IL20RA-mediated inhibition on OC metastasis, we firstly excluded the possibility that IL20RA might regulate the proliferation and migration of OC cells (*Figure 3—figure supplement 1A–D*). We next examined the possible impacts of IL20RA in shaping the immune microenvironment during the spreading of OC cells into the intraperitoneal cavity. In the syngeneic murine OC model by orthotopic transplant of ID8 cells, we analyzed the proportions of immune cells in ascites, including macrophages, T lymphocytes, and B lymphocytes. Compared with control ID8 cells with very low level of endogenous IL20RA (*Figure 1—figure supplement 1A*), reconstitution of IL20RA does not change the total proportion of macrophages (CD11b$^+$ F4/80$^+$) among leukocytes (CD45$^+$). However, we observed a significant increase in the proportion of M1-like (MHCII$^+$ CD206$^-$) macrophages and a dramatic decrease of M2-like (MHCII$^-$ CD206$^+$) macrophages in the malignant ascites caused by IL20RA-reconstituted ID8 cells (*Figure 3A, B*), which were further confirmed by dramatically increased M1-like marker genes and decreased M2-like markers (*Figure 3C*).

We also found that neither the ratios of T lymphocytes (CD45$^+$ CD3$^+$) and their subtypes (i.e., CD4$^+$ and CD8$^+$ T lymphocytes) nor that of B lymphocytes (CD45$^+$ B220$^+$) showed significant change upon the reconstitution of IL20RA (*Figure 3D, E*), which was consistent with the fact that IL20RA was screened in immunodeficient NOD-SCID mice lacking both T and B lymphocytes and NK cells (*Figure 1A*).

To further confirm that IL20RA in OC cells affects the macrophages during the transcoelomic metastasis, we examined the immune cells in malignant ascites caused by the inoculation of SK-OV-3 cells with silenced IL20RA into NOD-SCID mice. FACS results show that the population of M1-like (MHCII$^+$) macrophages is significantly decreased and M2-like (CD206$^+$) macrophages is significantly increased upon silencing IL20RA (*Figure 3F*). Collectively, these data demonstrate that IL20RA in OC cells is able to regulate the polarization of peritoneal macrophages, instead of T lymphocytes and B lymphocytes, to prevent the transcoelomic spreading of OC cells into the peritoneal cavity.

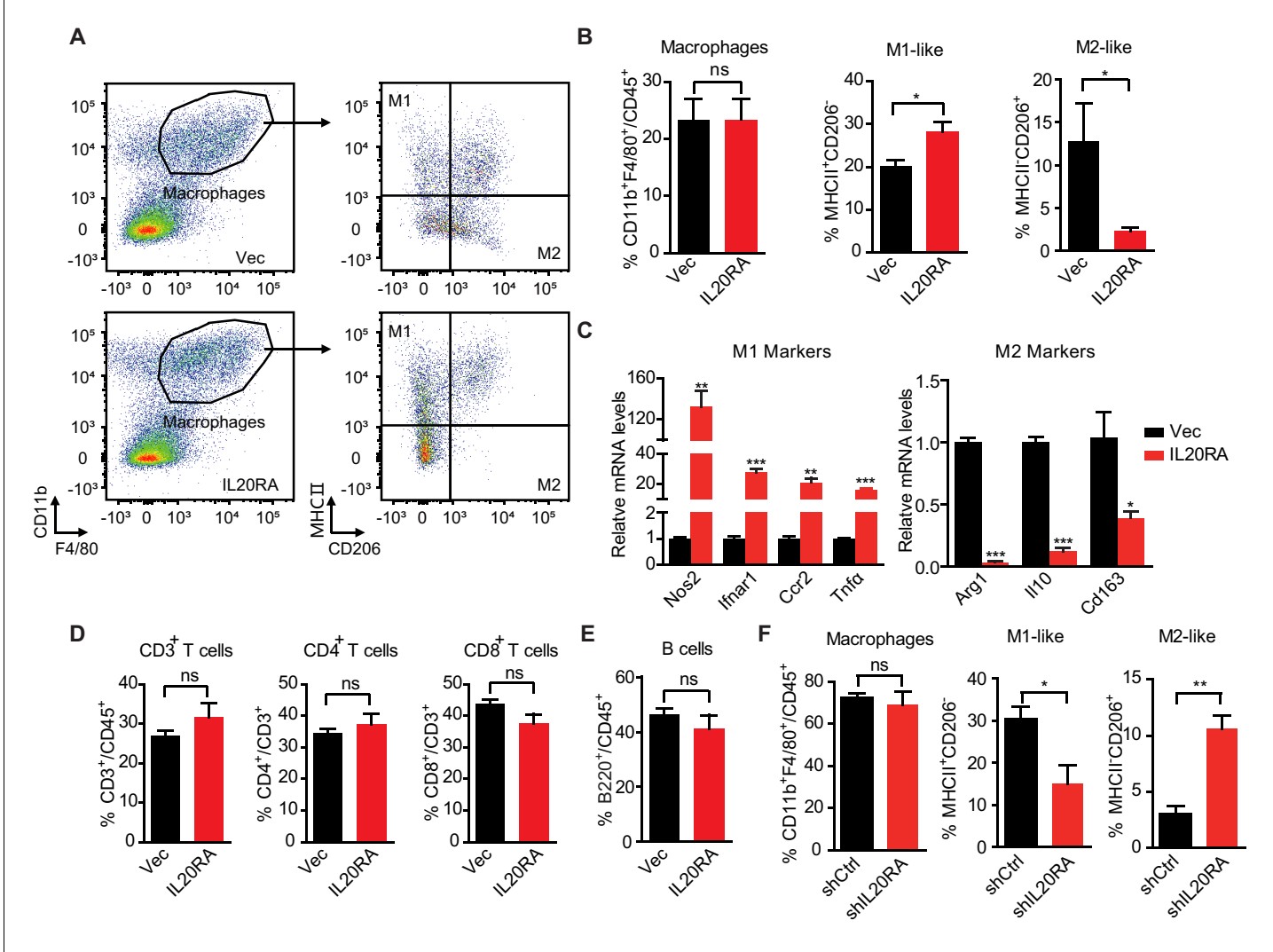

**Figure 3.** Characterization of IL20RA-mediated modulation on the peritoneal immune microenvironment during the transcoelomic metastasis of ovarian cancer (OC). (A, B) Flow cytometry analysis of macrophages (CD45$^+$ CD11b$^+$ F4/80$^+$) and M1-like (MHC II$^+$ CD206$^-$) and M2-like (MHC II$^-$ CD206$^+$) subpopulations in ascites formed in C57BL/6 mice at 60 days after orthotopically inoculated with IL20RA-reconsitituted or control (Vec) ID8 cells (A). The quantification is shown in (B) as means ± SEM (n = 5), *p<0.05, ns, not significant, by unpaired two-sided Student's t-test. (C) qRT-PCR analysis of key marker genes in peritoneal macrophages (CD11b$^+$ F4/80$^+$) isolated in (A) (shown as means ± SEM, *p<0.05, **p<0.01, ***p<0.001, by unpaired two-sided Student's t-test). (D, E) Flow cytometry analysis of peritoneal T lymphocytes (D) and B lymphocytes (E) from syngeneic OC mouse model (same mice as described in A), ns, not significant, by unpaired two-sided Student's t-test. (F) Flow cytometry analysis of macrophages in ascites formed in NOD-SCID mice at 40 days after orthotopically inoculated with indicated SK-OV-3 cells. Data are shown as means ± SEM, n = 3, *p<0.05, **p<0.01, ***p<0.001, by unpaired two-sided Student's t-test.

The online version of this article includes the following source data and figure supplement(s) for figure 3:

**Source data 1.** An Excel sheet with numerical quantification data.

**Figure supplement 1.** IL20RA has no effect on the proliferation and migration abilities of ovarian cancer (OC) cells.

## IL20RA mediates a direct conversation between OC cells and the macrophages that regulates the polarization of macrophages

To investigate if IL20RA supports a metastasis-preventing peritoneal immune-microenvironment through a direct crosstalk between OC cells and macrophages, we checked the polarization of macrophages in vitro under the stimulation of conditioned medium (CM) from OC cells (*Figure 4A*). CM from IL-20-stimulated IL20RA-reconstituted ID8 cells (hereafter referred to as 'CM$^{IL20RA\ IL-20\ (+)}$') significantly stimulates the expression of M1-like markers while decreases the expression of M2-like

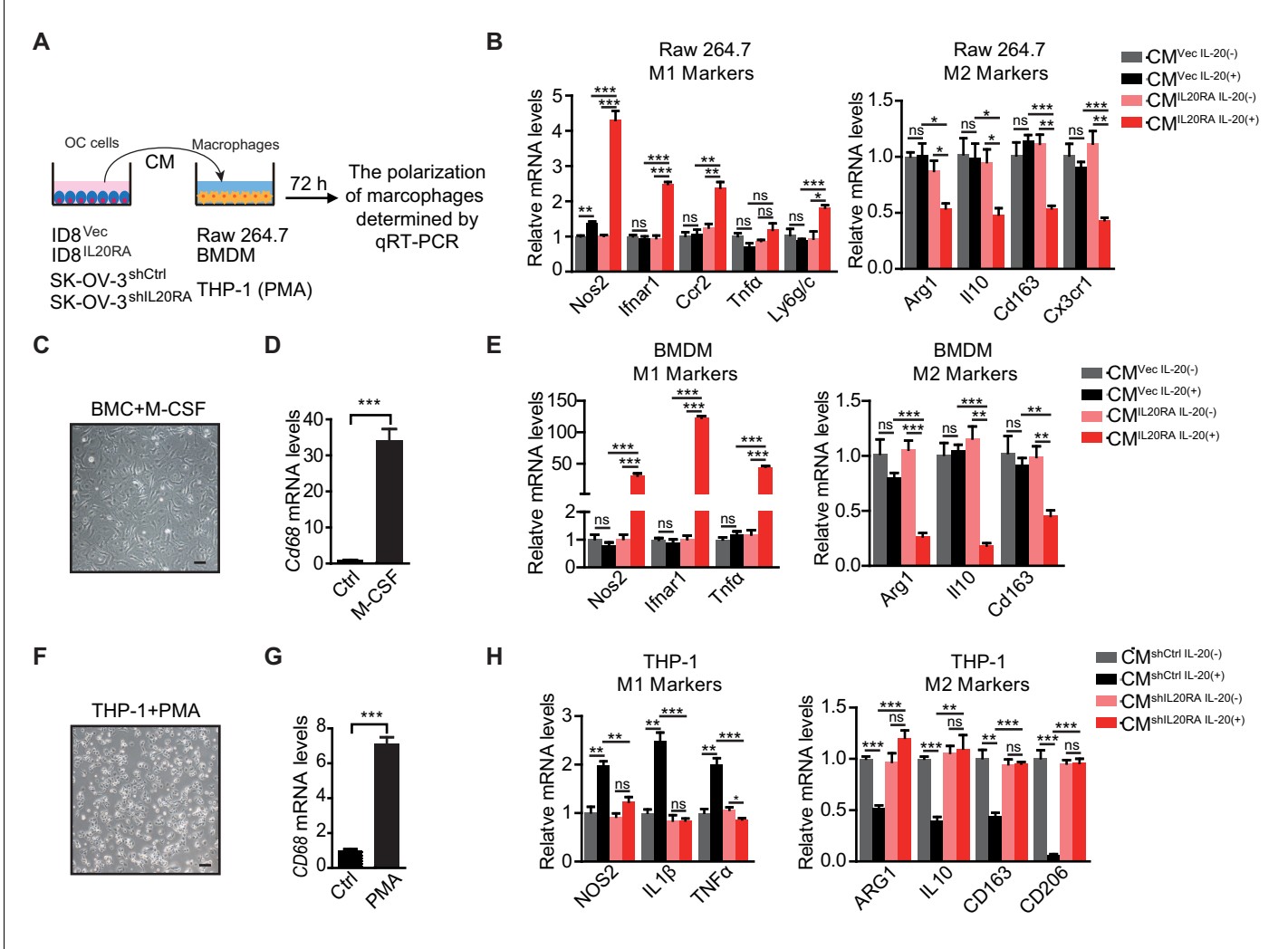

**Figure 4.** IL20RA mediates a direct crosstalk between ovarian cancer (OC) cells and macrophages to regulate the polarization of macrophages. (A) Schematic of the in vitro crosstalk experiment using conditioned medium (CM) from OC cells to educate macrophages. (B) qRT-PCR analysis of macrophage marker genes in RAW 264.7 cells stimulated with CM of IL-20-stimulated/unstimulated IL20RA-reconsitituted (CM$^{IL20RA}$) or control (CM$^{Vec}$) ID8 cells for 72 hr. (C, D) Representative image of bone marrow-derived macrophage (BMDM) differentiated from bone marrow cell with the treatment of macrophage colony stimulating factor for 7 days (C), which were further characterized by qRT-PCR analysis of its marker gene *Cd68* (D). Scale bar: 20 μm. (E) qRT-PCR analysis of macrophage marker genes in BMDM treated with CM of IL-20-stimulated/unstimulated IL20RA-reconsitituted (CM$^{IL20RA}$) or control (CM$^{Vec}$) ID8 cells for 72 hr. (F, G) Image of macrophages differentiated from THP-1 cells with the treatment of phorbol-12-myristate-13-acetate for 48 hr (F) and further characterization by qRT-PCR analysis of *CD68* (G). Scale bar: 20 μm. (H) qRT-PCR analysis of macrophage marker genes in THP-1-derived macrophages treated with CM from IL-20-stimulated/unstimulated shIL20RA or control shRNA (shCtrl)-transfected SK-OV-3 cells for 72 hr. All the qRT-PCR data are shown as means ± SEM from three independent experiments, *p<0.05, **p<0.01, ***p<0.001, by unpaired two-sided Student's t-test.

The online version of this article includes the following source data and figure supplement(s) for figure 4:

**Source data 1.** An Excel sheet with numerical quantification data.

**Figure supplement 1.** In vitro co-culture experiments to show the IL20RA-mediated crosstalk between ovarian cancer (OC) cells and macrophages for their polarization.

markers in RAW 264.7 cells when compared with CM from IL-20-stimulated empty vector-transfected ID8 control cells (referred to as CM$^{Vec\ IL-20\ (+)}$) (*Figure 4B*). To further confirm, we prepared bone marrow-derived macrophages (BMDMs) by treating the bone marrow cells (BMCs) isolated from healthy C57BL/6 mice with 20 ng/mL macrophage colony stimulating factor (M-CSF) for 7 days, which were characterized by the adherent morphology and the expression of *Cd68* (*Figure 4C, D*).

Consistently, CM[IL20RA IL-20 (+)] dramatically induces the expression of M1-like markers and decreases the M2-like markers in BMDM (*Figure 4E*).

In human macrophages differentiated from THP-1 monocytes by phorbol-12-myristate-13-acetate (PMA) treatment (*Figure 4F, G*), CM from IL-20-stimulated, IL20RA-silenced SK-OV-3 cells significantly increases the expression of M2-like markers and decreases the expression of M1-like markers when compared with CM from IL-20-stimulated control shRNA (shCtrl)-transfected SK-OV-3 cells (*Figure 4H*). Besides, the polarization of macrophages is not changed under the stimulation of CM from unstimulated IL20RA-reconstituted or IL20RA-silenced OC cells, indicating that IL20RA-mediated macrophage polarization needs the prior stimulation of the IL-20 (*Figure 4B, E, H*).

We also checked the polarization of macrophages in vitro using the indirect co-culture system (*Figure 4—figure supplement 1A*). IL-20-stimulated, IL20RA-reconstituted ID8 cells have no effect on the migration capacity of RAW 264.7 cells (*Figure 4—figure supplement 1B, C*). However, co-culture with IL-20-stimulated IL20RA-reconstituted ID8 cells dramatically induces the expression of M1-like markers and reduces the M2-like markers in RAW 264.7 cells (*Figure 4—figure supplement 1D*). Besides, co-culture with IL-20-stimulated, IL20RA-silenced SK-OV-3 cells significantly increases the expression of M2-like markers and decreases the expression of M1-like markers in THP-1-derived macrophages (*Figure 4—figure supplement 1E*). We also excluded the possibility that IL-20 and IL-24 directly regulated the polarization of macrophages since it did not affect the expression of both the M1- and M2-like markers in RAW 264.7 cells (*Figure 4—figure supplement 1F, G*). Collectively, these data reveal a conserved, IL20RA-mediated direct crosstalk between OC cells and macrophages that favors an inflammatory immune microenvironment.

## IL20RA-mediated education of macrophages plays essential roles in the prevention of the transcoelomic metastasis of OC

To investigate the function of the macrophages educated by IL20RA-mediated crosstalk with OC cells in the transcoelomic metastasis of OC, we injected ID8 cells alone or together with CM[Vec IL-20 (+)]- or CM[IL20RA IL-20 (+)]-educated BMDM into the peritoneal cavity of C57BL/6 mice (*Figure 5A*). CM[IL20RA IL-20 (+)]-educated BMDM significantly reduces the volume of malignant ascites (*Figure 5B, C*) and dramatically inhibits the formation of metastatic nodules in peritoneal cavity (*Figure 5D, E*), indicating that IL20RA-mediated crosstalk between OC cells and macrophages prevents the transcoelomic metastasis of OC. In addition, in peritoneal macrophage-depleted mice, reconstitution of IL20RA in ID8 cells can no longer reduce the volume of malignant ascites and inhibit the peritoneal metastasis of OC, indicating that IL20RA-mediated education of macrophages plays an essential role to suppress the transcoelomic metastasis of OC (*Figure 5F–J*).

## Mesothelial cells in peritoneum produce IL20RA ligands IL-20 and IL-24 when challenged with OC cells in the peritoneal cavity

To identify the signal sources in the peritoneal cavity that respond to the disseminated OC cells through IL20RA, we injected ID8 cells into the peritoneal cavity of C57BL/6 mice to mimic the peritoneal dissemination of OC cells and checked the expression of possible IL20RA ligands, including IL-19, IL-20, and IL-24, in different tissues in the peritoneal cavity. A dramatic increase of *Il20* and *Il24* from the abdominal wall occurs upon the injection of OC cells into the peritoneal cavity (*Figure 6A*), while there is no induction of these ligands in the intestine, ovary, and peritoneal macrophages (CD11b[+] F4/80[+]) (*Figure 6A, B*). In addition, reconstitution of IL20RA in ID8 cells does not induce these ligands in themselves (*Figure 6C*). The amounts of IL-20 and IL-24 in peritoneal flushing fluid are significantly increased upon the injection of OC cells into the peritoneal cavity (*Figure 6D*). To further confirm, we isolated abdominal wall and co-cultured with ID8 cells in vitro and a boosted expression of *Il20* and *Il24* was observed (*Figure 6E*). Hematoxylin and eosin (H&E) and IHC staining of the abdominal walls from mice bearing peritoneal-disseminated OC cells show that IL-20 and IL-24 are originated from mesothelial cells (*Figure 6F, G*), which are further confirmed by immunofluorescent (IF) staining (*Figure 6—figure supplement 1A, B*).

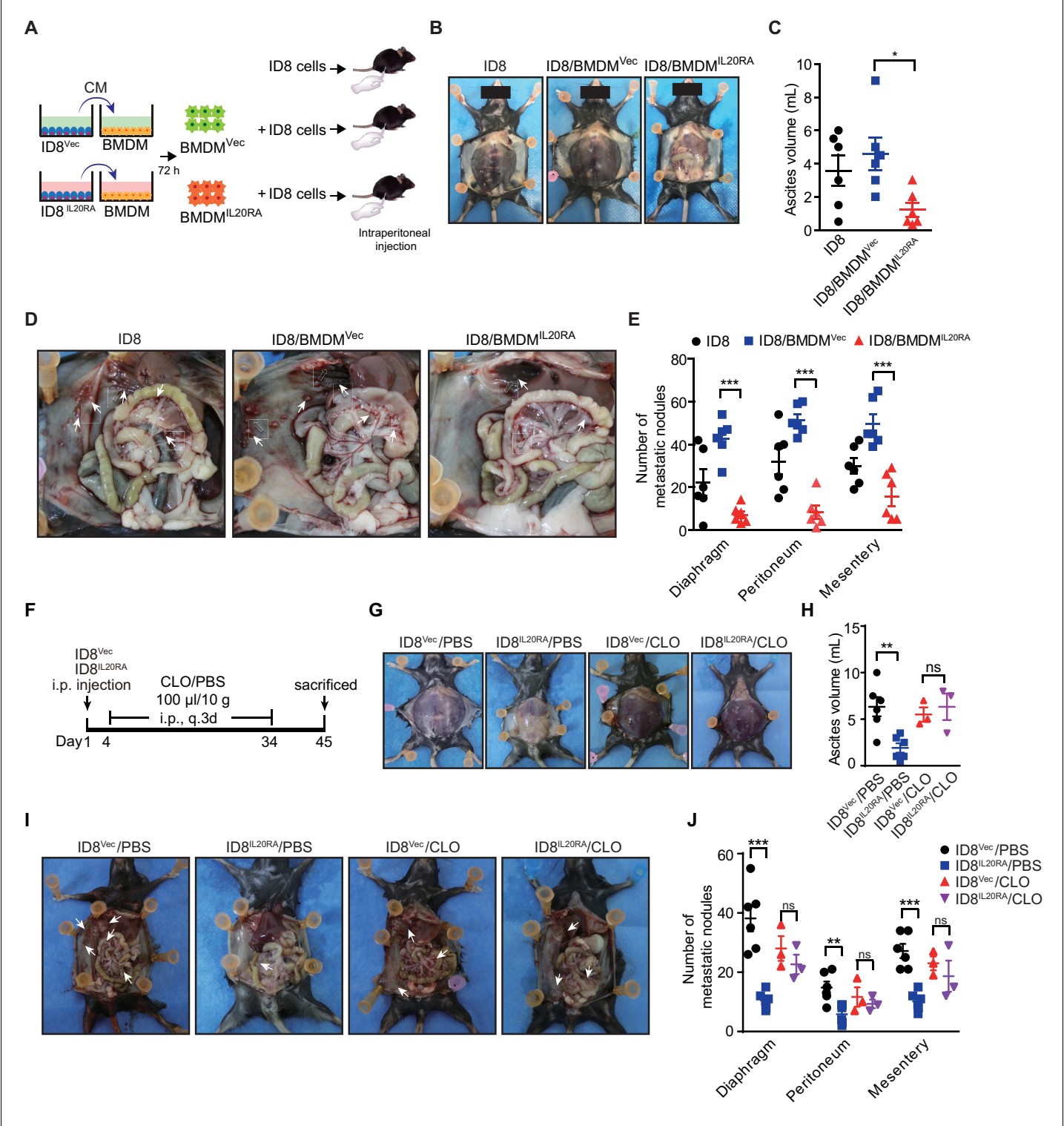

**Figure 5.** Macrophages play a prominent role in IL20RA-mediated suppression of ovarian cancer (OC) metastasis. (**A**) Schematic of the experiments. ID8 cells alone or mixed with CM$^{IL20RA\ IL-20\ (+)}$- or CM$^{Vec\ IL-20\ (+)}$-stimulated bone marrow-derived macrophage were injected into the peritoneal cavity of C57BL/6 mice. (**B–E**) Representative images of the ascites formation (**B**) and metastatic nodules in peritoneal cavity (**D**) at day 45 post-inoculation. The quantification of ascites and metastatic nodules is shown in (**C**) and (**E**), respectively. Data are shown as means ± SEM, n = 6, *p<0.05; **p<0.01; ***p<0.001, by unpaired two-sided Student's t-test. (**F**) Schematic of the experiments. IL20RA-reconsititued or control (Vec) ID8 cells were injected into the peritoneal cavity of C57BL/6 mice. The phosphate-buffered saline liposomes (PBS) or clodronate liposomes (CLO) were intraperitoneal (i.p.) injected every 3 days. (**G–J**) Representative images of the ascites formation (**G**) and metastatic nodules in peritoneal cavity (**I**) at day 45 post-inoculation. The

*Figure 5 continued on next page*

*Figure 5 continued*

quantification of ascites and metastatic nodules is shown in (**H**) and (**J**), respectively. Data are shown as means ± SEM, n = 6 (ID8^Vec^/PBS and ID8^IL20RA^/PBS), n = 3 (ID8^Vec^/CLO and ID8^IL20RA^/CLO), ns, not significant; **p<0.01; ***p<0.001, by unpaired two-sided Student's t-test.

The online version of this article includes the following source data for figure 5:

**Source data 1.** An Excel sheet with numerical quantification data.

## IL-20/IL20RA activates OAS/RNase L-mediated NLR signaling to produce mature IL-18 for macrophage polarization

To explore the IL-20/IL20RA-mediated downstream signaling that modulates the polarization of peritoneal macrophage, we analyzed the transcriptome of ID8 cells with reconstituted IL20RA under the stimulation of IL-20. Kyoto Encyclopedia of Genes and Genomes (KEGG) enrichment analysis shows that the differentially expressed genes are enriched in the immune system (*Figure 7—figure supplement 1A*). In particular, IL-20/IL20RA-mediated signaling greatly increases the expression of several genes, that is, 2′−5′-oligoadenylate synthetase (*Oas1a* and *Oas1g*) and *Il18*, involved in OAS-RNase L-mediated NOD-like receptor (NLR) signaling pathway (*Figure 7—figure supplement 1B, C*), which regulates inflammasome signaling to activate Caspase-1 and hence the production of functional inflammatory cytokines IL-1β and IL-18 by cleavage during viral infections (*Banerjee, 2016*; *Chakrabarti et al., 2015*).

The induction of *Oas1a* and *Il18* in IL20RA-reconsititued ID8 cells upon IL-20 stimulation is further confirmed by qRT-PCR (*Figure 7A*) and enzyme-linked immunosorbent assay (ELISA) for secreted IL-18 (*Figure 7B*). Western blot analysis shows that IL-20/IL20RA triggers the phosphorylation of STAT3 and results in increased OAS1A, activated Caspase-1 and IL-18 by cleavage (*Figure 7C*), which also occurs in human SK-OV-3 cells when stimulated by IL-20 (*Figure 7D, E*). We further identify a STAT3-binding site on the promoter of *Oas1a* gene by chromatin immunoprecipitation and qPCR (ChIP-qPCR) (*Figure 7F, G*), confirming that IL-20/IL20RA induces *Oas1a* and *Il18* through downstream STAT3. Consistently, IHC staining of human OC tissues shows that the levels of phosphorylated STAT3, OAS1, and IL-18 are significantly lower in metastatic lesions than those in primary sites (*Figure 7H, I*). In addition, transcriptome analysis on a cohort of 530 serous OC patients from TCGA database shows the positive correlation of *IL20RA* with both *OAS1* and *IL18* (*Figure 7—figure supplement 1D*), further supporting that IL-20/IL20RA activates OAS/RNase L-mediated inflammasome signaling in human OC as well.

To get insights into the role of IL-18 in macrophage polarization, RAW 264.7 cells were treated with IL-18, which resulted in greatly increased expression of M1-like markers and significantly decreased M2-like markers (*Figure 7J*). CM^IL20RA IL-20 (+)^-induced polarization of RAW 264.7 cells to M1-like phenotypes can be dramatically blocked by IL-18 neutralization antibody (*Figure 7K*).

To investigate the essential role of IL-18 in IL-20/IL20RA-mediated downstream signaling to suppress the peritoneal dissemination of OC cells in vivo, we knock down IL-18 in IL20RA-reconsititued ID8 cells (ID8^IL20RA/shIL-18^) and further inject ID8^Vec^, ID8^IL20RA^, and ID8^IL20RA/shIL-18^ cells into the peritoneal cavity of C57BL/6 mice (*Figure 8A, B*), which results in dramatically reduced IL-18 in the peritoneal fluids (*Figure 8C*). Silencing IL-18 abolishes the effects of reconstituted IL20RA in the inhibition of the malignant ascites formation and the peritoneal dissemination of OC cells (*Figure 8D–G*). In addition, IL-20/IL20RA-induced polarization of peritoneal macrophages to M1-like phenotypes is also disappeared upon the silenced of IL-18 (*Figure 8H*). These data suggest that IL-18 is the essential factor downstream IL-20/IL20RA signaling to regulate the polarization of macrophages and suppress the OC dissemination.

As IL20RB is the heterodimerization partner for IL20RA, we investigate the role of IL20RB in IL-20/IL20RA signaling-mediated OC metastasis. We knock down IL20RB in control ID8 cells and IL20RA-reconsititued ID8 cells (*Figure 8—figure supplement 1A, B*). Western blot analysis and qRT-PCR analysis show that silencing IL20RB blocks the activation of the STAT3/NLR signaling in IL20RA-reconstituted ID8 cells and hence the polarization of macrophages educated by ID8 cells (*Figure 8—figure supplement 1B, C*). In ID8 cell-grafted murine model of OC, silencing IL20RB in IL20RA-reconstituted ID8 cells dramatically abolishes the suppressive roles of IL20RA in the formation of malignant ascites and the peritoneal dissemination of OC cells, indicating that IL20RA

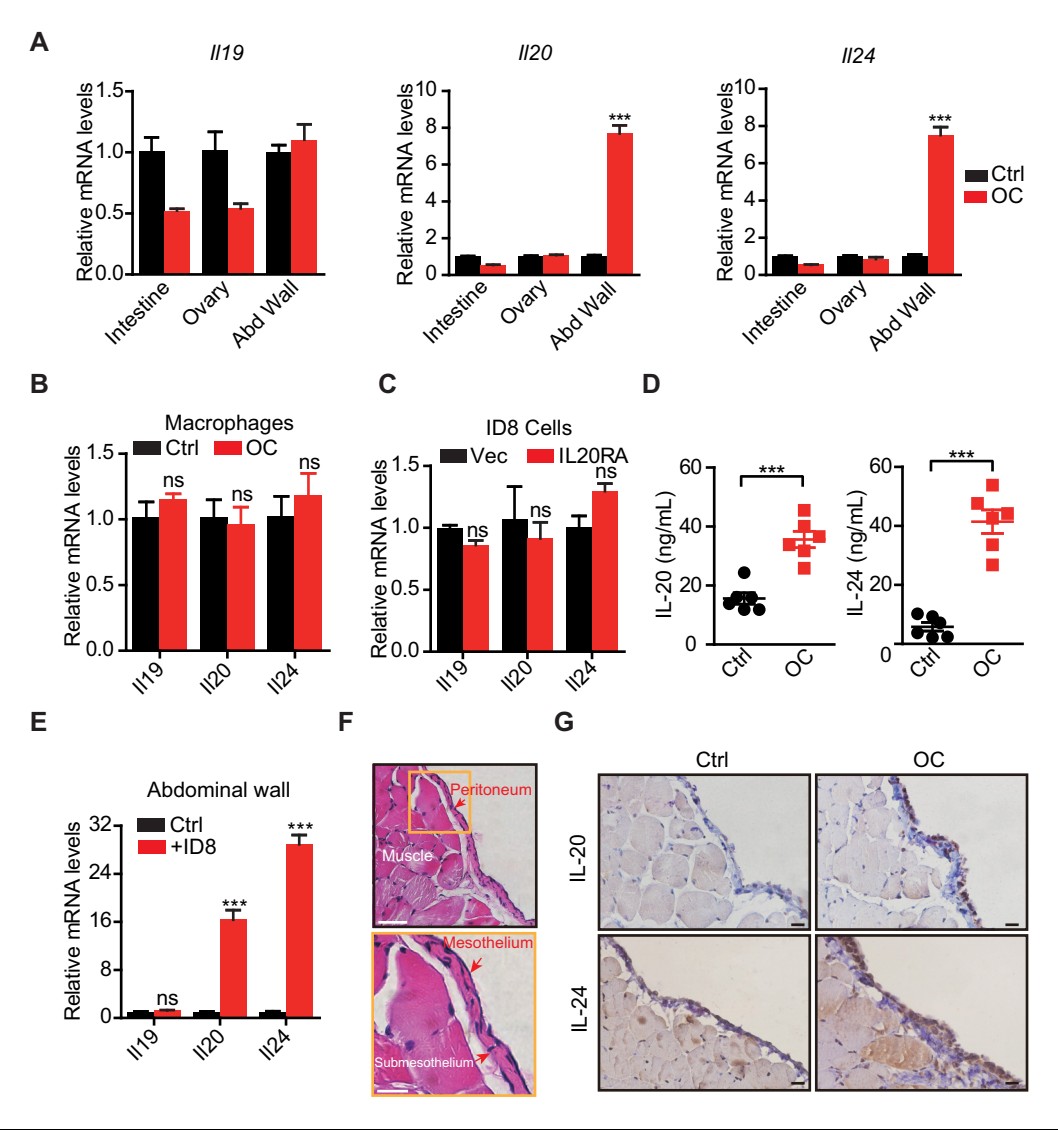

**Figure 6.** The mesothelial cells in peritoneum produce IL-20 and IL-24 when challenged by disseminated ovarian cancer (OC) cells in the peritoneal cavity. (A, B) qRT-PCR analysis of IL20RA ligands (*Il19*, *Il20,* and *Il24*) in peritoneal organs (intestinal, abdominal wall, ovary) (A) or peritoneal macrophages (CD11b[+] F4/80[+]) (B) taken from C57BL/6 mice with intraperitoneal injection of ID8 cells (OC) or phosphate-buffered saline (PBS) control (Ctrl) 9 days before. Data are shown as means ± SEM, n = 18 for Ctrl group and n = 6 for OC group, ***p<0.001, ns, not significant, by unpaired two-sided Student's t-test. (C) qRT-PCR analysis of IL20RA ligands in IL20RA-reconstituted or control (Vec) ID8 cells (means ± SEM, ns, not significant). (D) ELISA measurement of IL-20 and IL-24 in peritoneal flushing fluid from mice with intraperitoneal injection of ID8 cells (OC) or PBS control (Ctrl) 9 days before. n = 6 for each group. Data are shown as means ± SEM, ***p<0.001, by unpaired two-sided Student's t-test. (E) qRT-PCR analysis of the abdominal walls dissected from C57BL/6 mice and co-cultured with medium (Ctrl) or ID8 cells for 48 hr (means ± SEM from three independent experiments, ***p<0.001, ns, not significant, by unpaired two-sided Student's t-test). (F) Hematoxylin and eosin staining of the abdominal wall of C57BL/6 mice. Scale bar: 50 μm (upper panel); 20 μm (lower panel). (G) Immunohistochemical staining of IL-20 and IL-24 in abdominal walls dissected from mice with intraperitoneal injection of ID8 cells (OC) or PBS control (Ctrl) 9 days before. Scale bar: 20 μm.

The online version of this article includes the following source data and figure supplement(s) for figure 6:

**Source data 1.** An Excel sheet with numerical quantification data.
**Figure supplement 1.** Mesothelial cells in peritoneum are the major source of IL-20 and IL-24 when challenged by ovarian cancer (OC).

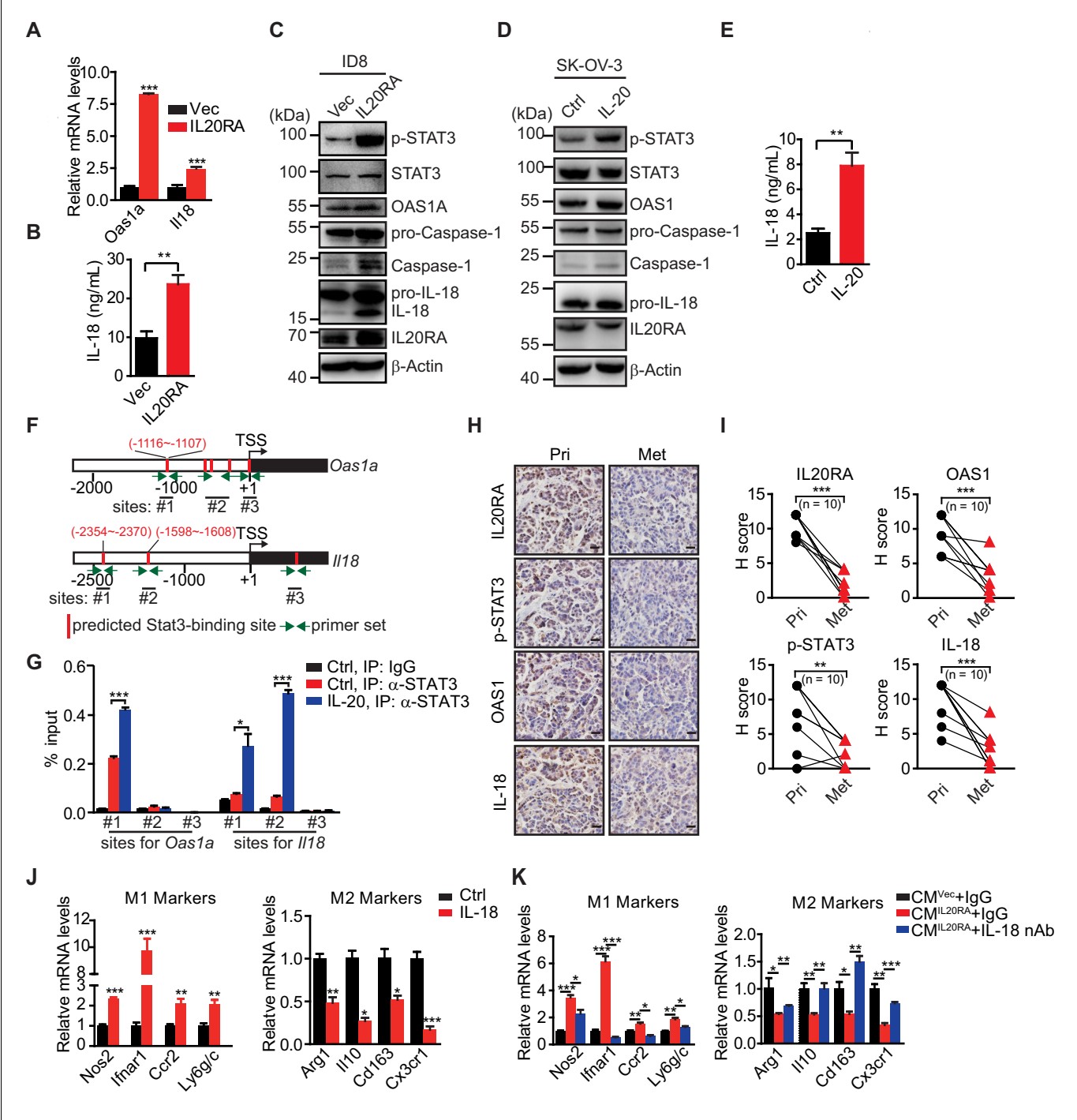

**Figure 7.** IL-20 activates the IL20RA-STAT3-OAS1/RNase L-NLR signaling to produce IL-18 to regulate the macrophages polarization. (**A**) qRT-PCR analysis of *Oas1a* and *Il18* upon IL20RA reconstitution in ID8 cells. (**B**) The secreted IL-18 from IL20RA-reconstituted and control ID8 cells was measured by ELISA. (**C**) Western blot analysis of indicated proteins in IL20RA-reconstituted and control ID8 cells under the stimulation of IL-20. (**D**) Western blot analysis of indicated proteins in SK-OV-3 cells stimulated with IL-20 or phosphate-buffered saline (PBS) (Ctrl) for 24 hr. (**E**) ELISA measurement of IL-18 secreted from SK-OV-3 cells stimulated by IL-20 for 24 hr. (**F**) Schematic of the *Oas1a* promoter and *Il18* promoter with predicted STAT3-binding sites and the primer sets. (**G**) IL20RA-reconstituted ID8 cells were stimulated with IL-20 or PBS (Ctrl) for 24 hr before the STAT3 binding on *Oas1a* promoter and *Il18* promoter was analyzed by ChIP-qPCR. (**H, I**) Immunohistochemical analysis of IL20RA, p-STAT3, OAS1, and IL-18 in human primary ovarian cancer tissues (Pri) and paired peritoneal metastatic nodules (Met) (**H**) and quantification (**I**). **p<0.01; ***p<0.001, by paired two-sided Student's t-test. Scale bar: 20 μm. (**J**) qRT-PCR analysis of macrophage marker genes in RAW 264.7 cells stimulated with IL-20 protein for 72 hr. (**K**) qRT-PCR analysis of macrophage marker genes in RAW 264.7 cells treated by CM$^{IL20RA\ IL-20\ (+)}$ or CM$^{Vec\ IL-20\ (+)}$ together with IL-18 neutralizing antibody (nAb) or nonspecific

*Figure 7 continued on next page*

*Figure 7 continued*

IgG (IgG) for 72 hr. All the qRT-PCR and ELISA data are shown as means ± SEM from three independent experiments, *p<0.05, **p<0.01, ***p<0.001, by unpaired two-sided Student's t-test.

The online version of this article includes the following source data and figure supplement(s) for figure 7:

**Source data 1.** An Excel sheet with numerical quantification data.
**Figure supplement 1.** Kyoto Encyclopedia of Genes and Genomes (KEGG) analysis shows that IL-20 activates the OAS1/RNase L-NLR/IL-18 signaling.

signaling-inhibited OC dissemination needs the functional IL20RA/IL20RB heterodimer receptor (*Figure 8—figure supplement 1D–G*).

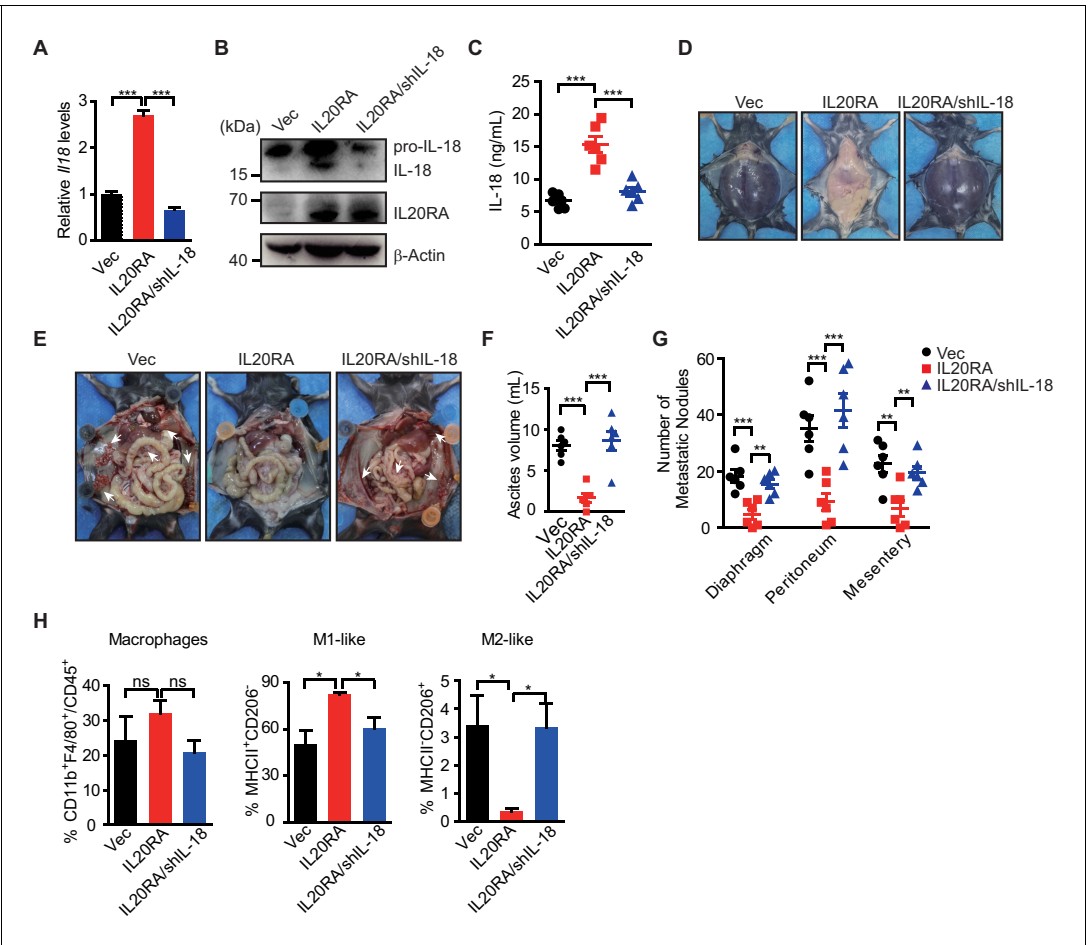

**Figure 8.** IL-18 is the essential factor downstream IL-20/IL20RA signaling to prevent the ovarian cancer dissemination. (A) qRT-PCR analysis of *Il18* in ID8$^{Vec}$ and ID8$^{IL20RA}$ cells transfected with shCtrl or shIL-18 (means ± SEM, ***p<0.001, by unpaired two-sided Student's t-test). (B) Western blot analysis of IL-18 in ID8$^{Vec}$ and ID8$^{IL20RA}$ cells transfected with shCtrl or shIL-18. (C) ELISA measurement of IL-18 in ascites from mice with intraperitoneal injection of ID8$^{Vec}$, ID8$^{IL20RA}$, and ID8$^{IL20RA/shIL-18}$ cells 45 days post-injection. Data are shown as means ± SEM, n = 6, ***p<0.001, by unpaired two-sided Student's t-test. (D–G) Representative images of ascites formation (D) and the metastatic nodules in peritoneal cavity (E) of C57BL/6 mice at day 45 after intraperitoneal injected with ID8$^{Vec}$, ID8$^{IL20RA}$, and ID8$^{IL20RA/shIL-18}$ cells. The quantification of ascites and metastatic nodules is shown in (F) and (G), respectively. Data are shown as means ± SEM, n = 6, **p<0.01, ***p<0.001, by unpaired two-sided Student's t-test. (H) Flow cytometry analysis of macrophages (CD45$^+$ CD11b$^+$ F4/80$^+$) and M1-like (MHC II$^+$ CD206$^-$) and M2-like (MHC II$^-$ CD206$^+$) subpopulations in ascites formed in C57BL/6 mice at day 45 after intraperitoneal injected with ID8$^{Vec}$, ID8$^{IL20RA}$, and ID8$^{IL20RA/shIL-18}$ cells. Data are shown as means ± SEM, n = 3, *p<0.05, ns, not significant, by unpaired two-sided Student's t-test.

The online version of this article includes the following source data and figure supplement(s) for figure 8:

**Source data 1.** An Excel sheet with numerical quantification data.
**Figure supplement 1.** IL20RA signaling-inhibited ovarian cancer dissemination needs the functional IL20RA/IL20RB heterodimer receptor.
**Figure supplement 2.** The cross-reaction of cytokines involved in IL20RA-mediated signaling.

As the role of IL-20/IL20RA in preventing the peritoneal dissemination of OC was observed in the murine models of OC established by both human and mouse OC cells (*Figure 1F–H*), we further investigate whether the cytokines involved in the process (IL-20, IL-24, and IL-18) have cross-reactivities between two species. Intraperitoneal injection of SK-OV-3 cells induces the increased expression of *Il20* and *Il24* from the abdominal wall of C57BL/6 mice (*Figure 8—figure supplement 2A*). Besides, a previous study has showed that murine IL-20 subfamily cytokines were also able to cross-react with human receptors (*Kolumam et al., 2017*). In addition, human IL-18 also significantly stimulates the expression of M1-like markers while inhibits the expression of M2-like markers in RAW 264.7 cells (*Figure 8—figure supplement 2B*). CM of the IL-20-stimulated IL20RA-silenced SK-OV-3 cells (CM^shIL20RA^) also can increase the expression of M2-like markers and inhibit the expression of M1-like markers of RAW 264.7 cells (*Figure 8—figure supplement 2C*). Therefore, these data suggest that IL-20/IL20RA signaling also works in NOD-SCID mice challenged with SK-OV-3 cells.

## Administration of IL-18 protein strongly suppresses the transcoelomic metastasis of OC

Given the dramatic decrease of IL20RA in the peritoneal metastasized OC cells (*Figure 2*) and the essential role of IL-18 as a major IL20RA downstream factor inhibiting peritoneal dissemination, we postulated that the direct administration of IL-18 might be an easy strategy to suppress the peritoneal growth of OC. To test, we treated the mice bearing orthotopically transplanted ID8 xenografts with recombinant IL-18 protein (*Figure 9A*). It shows that intraperitoneal injection of IL-18 dramatically reduces the ascites formation and the numbers of metastatic nodules in diaphragm, peritoneum, and mesentery (*Figure 9B–E*). Furthermore, direct administration of IL-18 significantly increases the proportion of M1-like (MHCII$^+$ CD206$^-$) macrophages and decreases M2-like (MHCII$^-$ CD206$^+$) macrophages in the malignant ascites (*Figure 9F*).

In conclusion, we discovered a new crosstalk of disseminated OC cells with peritoneal mesothelial cells and macrophages through IL-20/IL20RA/IL-18 axis in shaping the peritoneal immunomicroenvironment to prevent the transcoelomic metastasis of OC cells. OC cells, when disseminated into the peritoneal cavity, stimulate the mesothelial cells of the peritoneum to produce IL-20 and IL-24, which in turn activate the IL20RA downstream signaling in OC cells to trigger the OAS/RNase L-mediated NLR inflammasome signaling to produce mature IL-18, which consequently promotes the polarization of peritoneal macrophages into M1-like subtype to clear invaded OC cells in peritoneal cavity (*Figure 9G*). Highly metastatic OC cells always block this pathway by decreasing IL20RA expression, suggesting that reactivation of its downstream signaling, such as the application of IL-18, could be a useful strategy for the therapy of OC at advanced stages.

## Discussion

The common metastasis mode of OC is transcoelomic metastasis, in which the cancer cells disseminate into the peritoneal cavity and colonize on the peritoneum and abdominal organs. Transcoelomic metastasis occurs in about 70% of OC patients and many other types of cancers, such as pancreatic cancer, gastric cancer, endometrial cancer, and colon cancer (*Mikuła-Pietrasik et al., 2018*). Patients with transcoelomic metastasis are diagnosed at stages III and IV, which means high mortality and poor prognosis. The mechanisms, in particular, the peritoneal specific factors, behind this process still remain largely unclear. Here, through a genome-scale gene knockout screening in the orthotopic murine model of OC, we reveal an essential role of IL-20/IL20RA-mediated crosstalk between cancer cells and peritoneum mesothelial cells in regulating the polarization of peritoneal macrophages to prevent transcoelomic metastasis. This peritoneal-specific immune modulation mechanism may also happen in other types of cancers that can disseminate into the peritoneal cavity, such as gastric cancer, endometrial carcinoma, colon cancer, and bladder carcinoma, in which the high expression of IL20RA also correlates with a better clinical outcome of patients (*Figure 2—figure supplement 1*), highlighting the importance of IL-20/IL20RA-mediated epithelial immunity in preventing cancer metastasis into the peritoneal environment.

Immune cell-produced IL-20 subfamily cytokines and their receptors mainly expressed in epithelial cells facilitate the crosstalk between leukocytes and epithelial cells, which has essential functions in epithelial innate immunity for the host defense and tissue repair at epithelial surfaces. Due to the complicated effects of IL-20 subfamily cytokines in inflammation stimulation upon infection or injury

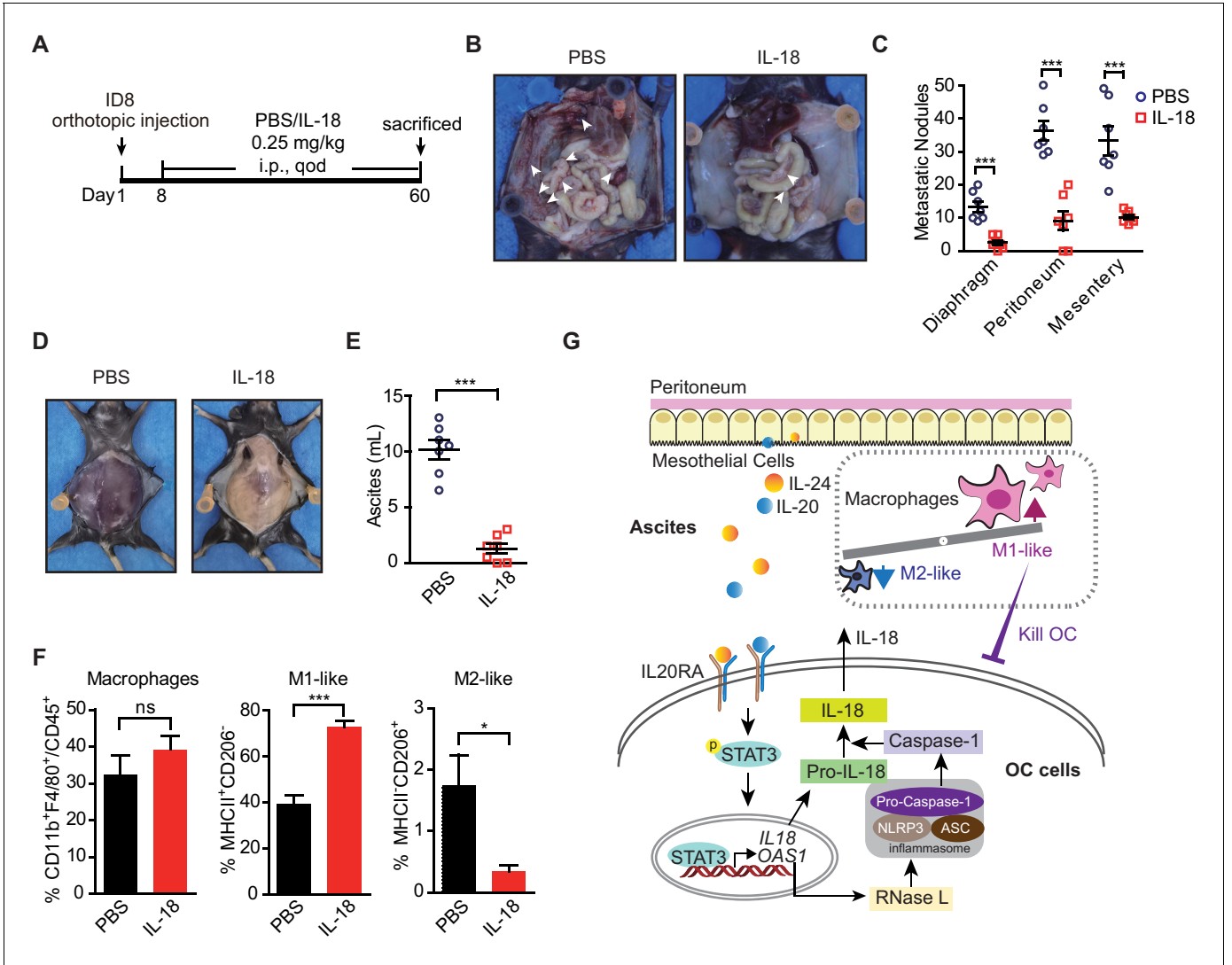

**Figure 9.** The therapeutic effect of recombinant IL-18 against the metastasis of ovarian cancer (OC). (A) Schematic of the experiments. ID8 cells were orthotopic injected into the ovaries of C57BL/6 mice. The phosphate-buffered saline or IL-18 protein were intraperitoneal (i.p.) injected every 2 days. (B–E) Representative images of the metastatic nodules in peritoneal cavity (B) and ascites formation (D) at day 60 post-inoculation. The quantification of metastatic nodules and ascites is shown in (C) and (E), respectively. Data are shown as means ± SEM (n = 7). ***p<0.001, by unpaired two-sided Student's t-test. (F) Flow cytometry analysis of macrophages (CD45⁺ CD11b⁺ F4/80⁺) and M1-like (MHC II⁺ CD206⁻) and M2-like (MHC II⁻ CD206⁺) subpopulations in ascites formed in C57BL/6 mice at 60 days after orthotopically inoculated with ID8 cells (means ± SEM, n = 5, *p<0.05, ***p<0.001, ns, not significant, by unpaired two-sided Student's t-test). (G) Schematics summarizing the IL-20/IL20RA-OAS1/RNase L-NLR-IL-18 axis in preventing the transcoelomic metastasis of OC.

The online version of this article includes the following source data for figure 9:

**Source data 1.** An Excel sheet with numerical quantification data.

and later immunosuppression for wound repair and tissue homeostasis restoration, both tumor-promoting and tumor-suppressing roles of IL-20 subfamily cytokines have been reported. IL-20 and IL-26 were discovered to promote proliferation and migration of bladder cancer, breast cancer, and gastric cancer cells, while IL-24 was found to inhibit the proliferation and metastasis of melanoma, prostate cancer, and OC (*Fisher et al., 2007*; *Gopalan et al., 2007*; *Hsu et al., 2012*; *Lee et al., 2013*; *Pradhan et al., 2018*; *You et al., 2013*). Although IL-20 subfamily cytokines have been extensively reported to activate STAT3 to exhibit oncogenic effects, STAT3 has also been shown to be able to switch its roles from tumor-promoting at early stages to tumor-suppressing in the invasion process (*Musteanu et al., 2010*). The different roles of IL-20 subfamily cytokines in tumor growth

and metastasis may depend on the local inflammatory environment that involves a complex network of conversations between cancer cells and various stromal components. For OC metastasized to omentum, cancer cell-mediated conversion of omentum stromal cells, including adipocytes, fibroblasts, macrophages, and mesenchymal stem cells, to cancer-associated stromal cells is of vital importance for the metastatic growth of OC cells on omentum (*Motohara et al., 2019*; *Thibault et al., 2014*). The abdominal cavity is lined with a layer of mesothelial cells that form the first line of defense against microbial pathogens through the expression of retinoic acid-inducible gene-I-like (RIG-I–like) receptors, Toll-like receptors, and C-type lectin-like receptors (*Colmont et al., 2011*; *Kato et al., 2004*; *Park et al., 2007*). Besides, mesothelial cells can secret cytokines, chemokines, and growth factors to regulate the inflammatory responses in peritoneal cavity (*Isaza-Restrepo et al., 2018*; *Yao et al., 2004*). However, whether and how mesothelial cells response to cancer cells invaded into the peritoneal cavity remain largely unknown. It is more difficult for OC cells to attach to mesothelial cells compared with the extracellular matrix and the fibroblasts (*Agarwal et al., 2019*; *Kenny et al., 2007*; *Niedbala et al., 1985*). During the metastatic growth, OC cells can destroy the mesothelial cell layer by promoting the mesothelial-to-mesenchymal transition (MMT) of mesothelial cells and hence the loss of cell-cell adherence (*Iwanicki et al., 2011*; *Kenny et al., 2011*; *Sandoval et al., 2013*). Here, we revealed, for the first time, that mesothelial cells can actively respond to OC cells invaded into the peritoneal cavity by secreting IL-20 and IL-24 to communicate with OC cells, consequently resulting in a further crosstalk between OC cells and macrophages for the formation of an antitumor microenvironment. This mesothelial-initiated defense mechanism is oftentimes overcome by successfully metastasized OC cells through the downregulation of IL-20/IL-24 receptor IL20RA (*Figure 2A–F*) to shut down the communication between OC and mesothelial cells.

It has been reported that in OC patients the majority of cell types in ascites are lymphocytes (37%) and macrophages (32%), contributing to the progression and metastasis of OC (*Kipps et al., 2013*). Generally, macrophages have the potential to differentiate into cancer-inhibiting M1-subtype and cancer-promoting M2-subtype. The ratio of M1/M2-subtype macrophages has predictive value for the prognosis of OC patients, with a higher M1/M2-subtype ratio being associated with a better prognosis, while a higher CD163$^+$ (M2-subtype)/CD68$^+$ (total macrophages) ratio with a poor prognosis (*Reinartz et al., 2014*; *Zhang et al., 2014*). Re-polarization of macrophages to increase the ratio of M1/M2 macrophages in ascites is a promising strategy for the treatment of OC. It has reported that host-produced histidine-rich glycoprotein (HRG) could inhibit the growth and metastasis of tumors via re-polarizing the macrophages from M2-subtype to M1-subtype (*Rolny et al., 2011*). Nanoparticles encapsulating miR-125b could specifically re-polarize the peritoneal macrophages to M1-subtype, thereby improving the efficacy of paclitaxel in OC (*Parayath et al., 2019*). Here, we reveal IL20RA as a key receptor in regulating the polarization of peritoneal macrophages, which is silenced in disseminated OC cells for the accumulation of M2-subtype macrophages in the peritoneal cavity for a successful metastatic growth. This indicates that reactivation of IL-20/IL20RA signaling by the application of IL-20 or IL-24 may not be a useful strategy to prevent OC transcoelomic metastasis. However, we also identify the key downstream effector of IL-20/IL20RA signaling that promotes M1-like macrophage, IL-18, which is usually involved in the regulation of innate immune responses against various pathogens (*Fabbi et al., 2015*).

IL-18 is an immunoregulatory cytokine with anti-cancer effect in many types of cancers (*Coughlin et al., 1998*; *Nishio et al., 2008*; *Salcedo et al., 2010*; *Zaki et al., 2010*). The normal ovarian and colon epithelia have the capacity to process and release active IL-18, thereby supporting a local antitumor immune microenvironment. However, during the malignant transformation, the ovarian and colon cancer cells lose the ability to process IL-18 as a result of the defective Caspase-1 activation (*Feng et al., 2005*; *Wang et al., 2002*). Thus, OC cells express more pro-IL-18 but poorly produce mature-IL-18. Elevated IL-18 was reported in the ascites and sera of OC patients in a 23 kDa pre-form, whereas the 18 kDa mature and active form was undetectable (*Orengo et al., 2011*). Here, we show that reconstitution of IL20RA in highly metastatic OC cells can reactivate the production of mature IL-18 via IL-20/IL20RA-OAS1/RNase L-NLR axis, suggesting the silence of IL20RA to be a possible reason for the loss of mature IL-18 in OC. IL-18 has been shown to exert antitumor activity through the activation of T cell (*Fabbi et al., 2015*). Here, our results expand its roles to promote the polarization of macrophages to inflammatory M1-like subtype to exhibit a potent antitumor effect against OC. Several clinical trials testing the antitumor efficacy of IL-18 have been

conducted in lymphomas and melanoma (*Robertson et al., 2013*; *Robertson et al., 2006*; *Tarhini et al., 2009*). Our study shows that the administration of recombinant IL-18 significantly suppresses the metastasis of IL20RA-deficient OC cells, suggesting the great value of IL-18 in the therapy of OC at advanced stages.

# Materials and methods

**Key resources table**

| Reagent type (species) or resource | Designation | Source or reference | Identifiers | Additional information |
|---|---|---|---|---|
| Gene (*Mus musculus*) | *Il20ra* | GenBank | Gene ID: 237313 | |
| Strain, strain background (*Escherichia coli*) | DH5α | Shanghai Weidi Biotechnology | Cat.#: DL1002 | |
| Strain, strain background (*Escherichia coli*) | BL21(DE3) | CWBIO | Cat.#: CW0809S | |
| Cell line (*Homo sapiens*) | THP-1 | ATCC | RRID:CVCL_0006 | |
| Cell line (*Homo sapiens*) | HEK 293T | ATCC | RRID:CVCL_0063 | |
| Cell line (*Homo sapiens*) | SK-OV-3 | ATCC | RRID:CVCL_0532 | |
| Cell line (*Mus musculus*) | RAW 264.7 | ATCC | RRID:CVCL_0493 | |
| Cell line (*Mus musculus*) | ID8 | Merck | RRID:CVCL_IU14 | |
| Transfected construct (*Homo sapiens*) | *IL20RA* shRNA | Sangon Biotech | | |
| Transfected construct (*Mus musculus*) | *Il18* shRNA | Sangon Biotech | | |
| Transfected construct (*Mus musculus*) | *Il20rb* shRNA | Sangon Biotech | | |
| Antibody | Anti-IL20RA (Rabbit polyclonal) | Abcam | RRID:AB_2123854 | WB (1:1000) IHC (1:200) |
| Antibody | Anti-IL20RB (Mouse monoclonal) | Proteintech | RRID:AB_11182930 | WB (1:1000) |
| Antibody | Anti-ACTB (Mouse monoclonal) | Immunoway | RRID:AB_2629465 | WB (1:5000) |
| Antibody | Anti-STAT3 (Mouse monoclonal) | Santa Cruz Biotechnology | RRID:AB_628293 | WB (1:1000) |
| Antibody | Anti-STAT3 (Mouse monoclonal) | Cell Signaling Technology | RRID:AB_331757 | ChIP (5 μL for 7 μg DNA) |
| Antibody | Anti-p-STAT3 (Mouse monoclonal) | Santa Cruz Biotechnology | RRID:AB_628292 | WB (1:1000) |
| Antibody | Anti-p-STAT3 (Rabbit polyclonal) | Signalway Antibody | RRID:AB_895980 | IHC (1:200) |
| Antibody | Anti-IL18 (Rabbit polyclonal) | Proteintech | RRID:AB_2123636 | WB (1:1000) IHC (1:200) |
| Antibody | Anti-OAS1A (Mouse monoclonal) | Santa Cruz Biotechnology | RRID:AB_10990911 | WB (1:1000) |
| Antibody | Anti-OAS1 (Rabbit polyclonal) | Proteintech | RRID:AB_2158292 | WB (1:1000) IHC (1:200) |
| Antibody | Anti-Caspase-1 (Rabbit polyclonal) | Proteintech | RRID:AB_2876874 | WB (1:1000) |
| Antibody | Anti-ARG1 (Rabbit polyclonal) | Proteintech | RRID:AB_2289842 | IHC (1:750) |
| Antibody | Anti-NOS2 (Rabbit polyclonal) | Proteintech | RRID:AB_2782960 | IHC (1:750) |
| Antibody | Anti-IL20 (Rabbit polyclonal) | Abclonal | RRID:AB_2750843 | IF (1:200) IHC (1:200) |

*Continued on next page*

Continued

| Reagent type (species) or resource | Designation | Source or reference | Identifiers | Additional information |
|---|---|---|---|---|
| Antibody | Anti-IL24 (Rabbit polyclonal) | Proteintech | RRID:AB_2880630 | IF (1:200) |
| | | | | IHC (1:200) |
| Antibody | Anti-Cytokeratin 18 (Mouse monoclonal) | Abcam | RRID:AB_305647 | IF (1:200) |
| Antibody | Anti-CD16/CD32 (Mouse monoclonal) | eBioscience | RRID:AB_467133 | FACS (1:200) |
| Antibody | Anti-CD45-PE (Mouse monoclonal) | Biolegend | RRID:AB_312971 | FACS (1:200) |
| Antibody | Anti-CD11b-APC (Mouse monoclonal) | Biolegend | RRID:AB_312795 | FACS (1:200) |
| Antibody | Anti-F4/80-APC/CY7 (Mouse monoclonal) | Biolegend | RRID:AB_10803170 | FACS (1:200) |
| Antibody | Anti-I-A/I-E-PERCP/CY5.5 (rat monoclonal) | Biolegend | RRID:AB_2191071 | FACS (1:200) |
| Antibody | Anti-CD206-FITC (Mouse monoclonal) | Biolegend | RRID:AB_10900988 | FACS (1:200) |
| Antibody | Anti-CD11c-FITC (Mouse monoclonal) | BD Biosciences | RRID:AB_396683 | FACS (1:200) |
| Antibody | Anti-CD3-FITC (Mouse monoclonal) | Biolegend | RRID:AB_312661 | FACS (1:200) |
| Antibody | Anti-CD4-APC (Rat monoclonal) | BD Biosciences | RRID:AB_398528 | FACS (1:200) |
| Antibody | Anti-CD8-PE/CY7 (Mouse monoclonal) | eBioscience | RRID:AB_469584 | FACS (1:200) |
| Antibody | Anti-CD45R/B220-FITC (Rat monoclonal) | BD Biosciences | RRID:AB_394617 | FACS (1:200) |
| Recombinant DNA reagent | pLV-H1-EF1α-puro (plasmid) | Biosettia | Cat.#: SORT-B19 | |
| Recombinant DNA reagent | pLV-EF1α-MCS-IRES-Bsd (plasmid) | Biosettia | Cat.#: cDNA-pLV03 | |
| Recombinant DNA reagent | pET-20b(+) vector (plasmid) | Novagen | Cat.#: 69739–3 | |
| Peptide, recombinant protein | Murine recombinant IL-20 protein | Origene | Cat.#: TP723795 | |
| Peptide, recombinant protein | Murine recombinant IL-24 protein | R&D Systems | Cat.#: 7807 ML-010 | |
| Peptide, recombinant protein | Human recombinant IL-20 protein | Sino Biological | Cat.#: 13060-HNAE | |
| Peptide, recombinant protein | Murine recombinant M-CSF | Peprotech | Cat.#: 315-02-10 | |
| Chemical compound | PMA | Sigma-Aldrich | Cat.#: 16561-29-8 | |
| Chemical compound | Clodronate liposomes | LIPOSOMA | Cat.#: CP-030030 | |
| Commercial assay or kit | Murine IL18 ELISA Kit | Wuxin Donglin Sci & Tech Development | Cat.#: DL-IL18-Mu | |
| Commercial assay or kit | Human IL18 ELISA Kit | Wuxin Donglin Sci & Tech Development | Cat.#: DL-IL18-Hu | |
| Commercial assay or kit | Murine IL20 ELISA Kit | Wuxin Donglin Sci & Tech Development | Cat.#: DL-IL20-Mu | |
| Commercial assay or kit | Murine IL24 ELISA Kit | Wuxin Donglin Sci & Tech Development | Cat.#: DLIL24-Mu | |
| Commercial assay or kit | ChIP-IT Express Kits | Active Motif | Cat.#: 53035 | |

## Cell culture and cytokine treatment

THP-1, RAW 264.7, A2780, SK-OV-3, and HEK 293T cells were purchased from the American Type Culture Collection (ATCC, Washington, USA). ES-2 cells were purchased from the Cell Bank of the Chinese Academy of Sciences (Shanghai, China). ID8 cells were purchased from Merck (Darmstadt, Germany) and screened in C57BL/6 mice for highly metastatic and HGSOC-like ID8 cells as described by Ward and colleagues (*Diaz Osterman et al., 2019*; *Ward et al., 2013*). Cell lines are mycoplasma free and genotypes were confirmed using Short Tandem Repeat (STR) profiling. SK-OV-3 and OVCAR-5 cells were cultured in McCoy's 5A Medium Modified (Biological Industries, Israel). ES-2, A2780, ID8, and HEK 293T cells were cultured in Dulbecco's modified Eagle's medium (DMEM) with high glucose (Biological Industries). THP-1 and RAW 264.7 cells were cultured in RPMI-1640 (Biological Industries). All the mediums were supplemented with 10% fetal bovine serum (Biological Industries), 100 U/mL penicillin, and 100 µg/mL streptomycin (Gibco, Grand Island, NY) and maintained at an atmosphere of 5% $CO_2$% and 95% air at 37°C.

To study the role of IL-18, IL-20, and IL-24 protein in the polarization of RAW 264.7 cells, RAW 264.7 cells were treated with 5 ng/mL murine recombinant IL-18 protein or 5 ng/mL human recombinant IL-18 protein (10119-HNCE, Sino Biological, Beijing, China) or 5 ng/mL murine recombinant IL-20 protein (TP723795, Origene, Rockville, MD) or 5 ng/mL murine recombinant IL-24 protein (7807 ML-010, R&D Systems, Minneapolis, MN) for 72 hr, respectively. To detect the activation of signaling pathway, ID8 and SK-OV-3 cells were stimulated with 2.5 ng/mL murine and human recombinant IL-20 protein (Sino Biological) for 24 hr, respectively.

## The differentiation of THP-1 and BMCs

THP-1 was differentiated into macrophages by the treatment with 100 ng/mL PMA (Sigma-Aldrich, St Louis, MO) for 48 hr. BMCs were isolated from the femurs and tibias of C57BL/6 mice. The bone marrow in the femurs and tibias were flushed and collected with DMEM/F-12. Erythrocytes in bone marrow were lysed using Red Blood Cell Lysis Buffer (Solarbio, Beijing, China). BMCs were cultured in DMEM/F-12 supplemented with 20 ng/mL murine M-CSF (315-02-10, Peprotech, Cranbury, NJ) in a density of $4 \times 10^6$/mL. The medium was changed every 3 days. BMCs were differentiated into BMDM with the stimulation of M-CSF for 7 days. 5 mM of EDTA was used to detach the cells from the culture dishes. The detached cells were seeded in the six-well plates for subsequent treatment with CM from cancer cells.

## Establishment of stable cell lines

Knocking down IL20RA, IL-18, and IL20RB was achieved by the insertion of shRNA templates into the lentivirus- based RNAi vector pLV-H1-EF1α-puro (Biosettia, San Diego, CA). The sequence of shRNA targeting human *IL20RA* (shIL20RA) is 5′ AAAAC CCTCT CCTGT AAGAA CAAGT TGGAT CCAAC TTGTT CTTAC AGGAG AGGGT 3′, murine *Il18* (shIL-18) is 5′ AAAAC CCTCT CCTGT AAGAA CAAGT TGGAT CCAAC TTGTT CTTAC AGGAG AGGGT 3′, murine *Il20rb* (shIL20RB) is 5′ AAAAG CTGGC ACTAG CTCTG TTTGC TTGGA TCCAA GCAAA CAGAG CTAGT GCCAG C 3′. The control shRNA plasmid targeting *Escherichia coli lacZ* gene (shCtrl) was purchased from Biosettia with the sequence of 5′ AAAAG CAGTT ATCTG GAAGA TCAGG TTGGA TCAAC CTGAT CTTCC AGATA ACTGC 3′. The coding sequence of mouse *Il20ra* was synthesized by Sangon Biotech (Shanghai, China) and cloned into plasmid pLV-EF1α-MCS-IRES-Bsd (Biosettia). All plasmids were validated by sequencing. The procedures of lentivirus package and infection were described previously (*Chang et al., 2015*). Cells infected with lentivirus for RNAi or reconstituted expression were selected with 2.5 µg/mL of puromycin and 2.5 µg/mL of blasticidin (Thermo Fisher Scientific, Waltham, MA), respectively.

## CRISPR/Cas9 knockout in vivo screening

GeCKO v2 human library made by Zhangfeng's lab was purchased from Addgene (Watertown, MA) amplified as described (*Joung et al., 2017*). The library contains 122,756 sgRNAs targeting 19,050 genes and 1864 miRNAs, and 1000 non-targeting sgRNAs. Lentivirus carrying the library were generated and infected SK-OV-3 cells with the multiplicity of infection (MOI) around 0.5. After the selection with puromycin, 2 million SK-OV-3 cells expressing sgRNAs were orthotopically injected into NOD-SCID mice. The mice were sacrificed after 40 days. Primary tumors were dissected. Metastatic

nodules in the peritoneal cavity were isolated, in vitro expanded for next run of orthotopic injection. After three rounds of in vivo selection, peritoneal metastatic cells were collected for subsequent high-throughput DNA deep sequencing to identify metastasis-related candidate genes. The RIGER P analysis was used to analyze the data of sgRNA libraries sequencing.

## Murine OC models

All of the animals were handled according to approved Institutional Animal Care and Use Committee protocols of the Nankai University. Animal experiments were approved by Institutional Animal Care and Use Committee of Nankai University (ethic approved number: 20180014). All attempts were made to minimize the handling time during surgery and treatment so as not to unduly stress the animals. Animals are observed daily after surgery to ensure there are no unexpected complications. 6–8-week-old female NOD-SCID and C57BL/6 mice were purchased from SPF Biotechnology (Beijing, China). 2 million human SK-OV-3 cells or 5 million murine ID8 cells suspended in 20 µL phosphate-buffered saline (PBS) were orthotopically injected into the left ovary of NOD-SCID and C57BL/6 mice, respectively. Ascites volumes and the numbers of metastatic nodules were measured, and total cells in ascites were harvested 40 days post-inoculation for the SK-OV-3 OC xenograft model and 60 days post-inoculation for the ID8 OC syngeneic model. For the intraperitoneal OC mouse model, 5 million ID8 cells suspended in 100 µL of PBS were injected directly into the peritoneal cavity of C57BL/6 mice. At day 45 post-injection, ascites and the number of metastatic nodules were analyzed.

To investigate the role of peritoneal macrophage in IL20RA-mediated OC metastasis, 5 million ID8 cells alone or together with $1.25 \times 10^6$ of BMDM (educated by CM from ID8 cells for 72 hr) in 100 µL of PBS were directly injected into the peritoneal cavity of C57BL/6 mice. In another model, 5 million IL20RA-reconstituted or control ID8 cells were injected into the peritoneal cavity of C57BL/6 mice. The mice were then intraperitoneal injected with PBS liposomes or clodronate liposomes (100 µL/10 g of mouse weight) (CP-030030, LIPOSOMA, Netherlands) every 3 days from day 4 to day 34 post-injection.

To identify the source of IL20RA ligands in peritoneal cavity, 5 million ID8 cells in 100 µL PBS (OC) or only 100 µL PBS (control) were directly injected into the peritoneal cavity of C57BL/6 mice. To investigate whether the intraperitoneal injection of SK-OV-3 cells was able to stimulate the production of IL20RA ligands, 5 million SK-OV-3 cells in 100 µL PBS (OC) or only 100 µL PBS (control) were directly injected into the peritoneal cavity of C57BL/6 mice. The mice were sacrificed 9 days post-injection. Peritoneal flushing fluid by PBS was collected. Total cells in peritoneal cavity were harvested for sorting the peritoneal macrophages. The total RNAs from intestine, abdominal wall, and ovary were extracted by using TRIzol reagent (Thermo Fisher Scientific) for subsequent gene expression analysis.

To investigate the therapeutic effect of IL-18, 5 million murine ID8 cells suspended in 20 µL PBS were orthotopically injected into the left ovaries of C57BL/6 mice. The mice were intraperitoneal injected with PBS or IL-18 protein (0.25 mg/kg of mouse weight) every 2 days from day 8 to day 60 post-injection.

## Analysis of OC patient samples

OC specimens from human patients were obtained from the Tianjin Center Hospital of Gynecology Obstetrics. The patients' study was performed in accordance with the ethics committee of Nankai University and Tianjin Center Hospital of Gynecology Obstetrics (ethic approved number: 2018KY032) and conformed to the principles embodied in the Declaration of Helsinki. All OC patients from the Tianjin Center Hospital of Gynecology Obstetrics provided informed consent. Twenty fresh paired samples, including primary serous OC tissues, ascites, and metastasis nodules in peritoneal cavity, were obtained from OC patients during operation between 2017 and 2019. The inclusion criteria included serous ovarian adenocarcinoma, age >40, and TNM stage III–IV. Primary tumor tissues and metastases samples were dissected into two parts: one stored in 4% paraformaldehyde for IHC staining, the other stored in TRIzol reagent for the extraction of protein and RNA. Cancer cells in ascites were harvested for the extraction of protein and RNA. The histopathology and TNM stages of OC patients were confirmed by pathologists and clinical doctors.

The analysis on the correlation between the prognosis of OC patients and the expression of IL20RA or IL20RB were performed using the online Kaplan–Meier plotter tool (http://www.kmplot.com/) on serous OC cohorts (*Gyorffy et al., 2012*). The co-expression analysis on IL20RA with IL18 and OAS1 in OC patients was examined using the data from TCGA database (Ovarian Serous Cystadenocarcinoma, Provisional, n = 530). We used mRNA RNA-seq on pan-cancer section in the online Kaplan–Meier plotter tool to analyze the correlation between the prognosis of other cancer types and the expression of IL20RA.

## Protein extraction and western blot

The tissues were homogenized by TissueLyser (SCIENTZ-48, Ningbo, Zhejiang, China). Total protein in tissues was extracted using TRIzol reagent. Cells were lysed in RIPA lysis buffer with protease inhibitor cocktail (Roche Life Science, Switzerland) and phosphatase inhibitor cocktail (Sigma-Aldrich). The concentrations of proteins were measured by Pierce BCA Protein Assay Kit (Thermo Fisher Scientific), and 20 μg of total protein were loaded into Tris-acrylamide gels. Proteins were transferred onto PVDF membranes (Merck). The membranes were blocked with 5% defatted milk before blotted with primary antibodies and secondary antibodies (anti-mouse IgG-HRP [at 1:2000 dilution], anti-rabbit IgG-HRP [at 1:2000 dilution] [ZSGB-BIO, Beijing, China]). The membranes were incubated with Immobilon Western Chemiluminescent HRP Substrate (Merck) and visualized and photographed by Tanon-5200 (Tanon, Shanghai, China).

## IL-18 protein expression and purification

The coding sequence of the mature IL-18 was cloned into pET-20b plasmid (Novagen) before being transduced into *E. coli* BL21 (DE3) cells (CWBIO, Beijing, China). The expression of recombinant IL-18 was induced by 0.5 mM IPTG (Thermo Fisher Scientific) for 8 hr at 37°C. Cells were harvested and lysed by sonication in the lysis buffer (300 mM NaCl, 50 mM $NaH_2PO_4$ pH 8.0, 10 mM imidazole, and 2 mg/mL lysozyme). The supernatants were incubated with Ni-NTA agarose beads (Qiagen) overnight at 4°C. The beads were consecutively washed with wash buffer containing 20 mM imidazole (300 mM NaCl, 50 mM $NaH_2PO_4$ pH 8.0, 20 mM imidazole). The recombinant proteins were eventually eluted with elution buffer containing 250 mM imidazole (300 mM NaCl, 50 mM $NaH_2PO_4$ pH 8.0, 250 mM imidazole) and were dialyzed to remove imidazole. The endotoxin in the purified protein was removed using High Performance Endotoxin Removal Agarose Resin (20518ES10, Yeasen Biotech, Shanghai, China) and examined by using Kinetic Turbidimetric LAL (60401ES06, Yeasen Biotech) (final concentration of endotoxin <0.03 EU/mL).

## qRT-PCR

1 μg of total RNAs from cells and tissues were reversely transcribed into cDNAs using M-MLV reverse transcriptase (Promega, Madison, WI). Quantitative PCR analysis was performed by using SYBR Green SuperMix (Yeasen Biotech) on LightCycler96 system (Roche) with the program as follows: 95°C for 300 s, 45 cycles of two-step reaction, that is, 95°C for 30 s followed by 60°C for 45 s. The primer sequences are listed in *Supplementary file 1*. The relative gene expression was calculated by the $2^{-\Delta\Delta Ct}$ method with β-actin used for the normalization.

## H&E staining, IHC, and IF

The tissues dissected from patients and mice were fixed in 4% paraformaldehyde, dehydrated, paraffin-embedded, consecutive sectioned in 6 μm thickness, and then deparaffinized. For H&E staining, tissue sections were stained with hematoxylin and eosin (Origene). For IHC staining, after treated with antigen retrieval solution (Solarbio) and 3% hydrogen peroxide, tissue sections were blocked with 5% goat serum, incubated with primary antibodies, biotin-conjugated secondary antibodies (Vector Laboratories, Burlingame, CA) and streptavidin-HRP (Vector Laboratories). The sections were visualized with 3,3′-diaminobenzidine (DAB) substrate (ZSGB-BIO), then counterstained with hematoxylin. Images were taken using an Olympus BX53 microscope (Tokyo, Japan). H score was used to quantify the degree of immunostaining, which was calculated by multiplying positively stained area (P) with staining intensity (I). The degrees for P were 0–4 (0, <5%; 1, 5–25%; 2, 25–50%; 3, 50–75%; 4, 75–100%). The degrees for I were 0–3 (0, none; 1, weak; 2, moderate; 3, strong). For IF staining, tissue sections were blocked with 5% goat serum, incubated with primary antibodies,

secondary antibodies conjugated with Alexa Fluor-488 and Alexa Fluor-594 (Thermo Fisher Scientific), then counterstained with 4′,6-diamidino-2-phenylindole (DAPI, Sigma-Aldrich). Images were taken using an Olympus FV1000 confocal microscope (Tokyo, Japan).

### Flow cytometry

Total ascites was collected when the mice were sacrificed. Erythrocytes in ascites were lysed using Red Blood Cells Lysis Buffer (Solarbio). Cells were washed using PBS with 1% FBS and passed through a 40 µm nylon filter (BD Biosciences, San Diego, CA, USA), then cells were Fc-blocked with anti-CD16/32, and stained with CD45-PE, CD11b-APC, and F4/80-APC/CY7 antibodies for detecting macrophages. MHC II-PERCP/Cy5.5 and CD206-FITC antibodies were used to differentiate M1- and M2-subtype macrophages. For detecting T cells and B cells, cells were stained with CD45-PE, CD3-FITC, CD4-APC, CD8-PE/CY7, and CD45R/B220-FITC antibodies. Cells were incubated with the antibodies for 30 min in the dark and washed twice using PBS with 1% FBS. LSRFORTESSA (BD Biosciences) was used to obtained the data, and FlowJo software (BD Biosciences) was used to analyze the data.

### Isolation of peritoneal macrophages

Total cells in peritoneal cavity of healthy C57BL/6 mice were harvested by peritoneal lavage using PBS. As per the staining protocol described in flow cytometry, cells from peritoneal flushing fluid of healthy mice and ascites of OC mice were stained with CD11b-APC and F4/80-APC/CY7 antibodies. CD11b$^+$ F4/80$^+$ peritoneal macrophages were isolated from total cells using FACSAria SORP (BD Biosciences) and processed for RNA extraction.

### Cell proliferation and migration assay

The proliferation of ID8 and SK-OV-3 cells was examined by cell counting under the stimulation of IL-20. 2000 ID8 cells and SK-OV-3 cells were seeded in 96-well plates evenly. The cells were digested and counted by using the Invitrogen Countess II FL Automated Cell Counter (Thermo Fisher Scientific) every 24 hr until they reached nearly 100% confluence.

Transwell Permeable Supports System (Corning Inc, Corning, NY, USA) was used for SK-OV-3 cells migration assays. $2 \times 10^4$ cells in 200 µL medium containing 2% FBS were seeded into the top chamber of an 8 µm Millipore transwell insert in a 24-well plate, and medium containing 10% FBS was added to the bottom chamber. After 8 hr, cells migrated to the bottom surface of the transwell membranes were fixed in 4% paraformaldehyde and stained with crystal violet (Beyotime, Shanghai, China). The migration ability of RAW 264.7 cells was examined using indirect transwell cell culture system (8 µm, Millipore). $4 \times 10^5$ of ID8 cells were seeded in six-well plates evenly. After 24 hr, $1 \times 10^5$ of RAW 264.7 cells were seeded in the upper transwell chamber. After co-cultured for 24 hr, cells migrated to the bottom surface of the transwell membranes were fixed in 4% paraformaldehyde and stained with crystal violet. The membranes were washed by PBS. Images of the membranes were taken using an Olympus BX53 microscope (Tokyo, Japan), and the numbers of migrated cells were counted in five randomly chosen fields.

### CM stimulation and co-culture system

IL20RA-silenced SK-OV-3, IL20RA-reconstituted ID8 or their parallel control cells were cultured in 10 cm dishes for 72 hr with/without the stimulation of IL-20 protein, and their CM were collected and centrifuged at 2000 rpm for 10 min to obtain the supernatants. $2 \times 10^5$ of RAW 264.7 cells, $1 \times 10^6$ of BMDM, and $1 \times 10^6$ of THP-1-derived macrophages were seeded in six-well plates 24 hr before the medium was changed to fresh medium mixed with OC cell-derived CM at the ratio of 2:1 and incubated for 72 hr. The macrophages were then collected in TRIzol reagent for RNA extraction. To investigate the role of IL-18 in regulating the polarization of macrophages, RAW 264.7 cells were stimulated with the ID8 cell-derived CM supplemented with 5 µg/mL of IL-18 neutralizing antibody (R&D Systems) or 5 µg/mL of rat IgG1 Isotype as a control (Thermo Fisher Scientific).

Macrophages and OC cells were indirect co-cultured using 0.4 µm pore membrane transwell (Merck). $2 \times 10^5$ of RAW 264.7 cells and $1 \times 10^6$ of THP-1-derived macrophages were seeded in six-well plates and cultured for 24 hr before $1 \times 10^5$ of SK-OV-3 cells or $5 \times 10^4$ of ID8 cells stimulated with IL-20 protein were seeded in the upper transwell chamber. After co-cultured for 72 hr,

macrophages were collected in TRIzol reagent for subsequent gene expression analysis. To investigate whether OC cells induce the production of IL-20 and IL-24 from peritoneum in vitro, abdominal wall and OC cells were co-cultured. $2 \times 10^5$ of ID8 cells were seeded in six-well plates. After 24 hr, $2 \times 2$ cm$^2$ abdominal wall tissue collected from C57BL/6 mice was put in the medium of ID8 cells and co-cultured for 48 hr.

## ELISA

Cultured medium from ID8 and SK-OV-3 cells were collected. Total protein of cultured cells was extracted by using the cell lysis buffer (20 mM Tris-HCl [pH 7.5], 150 mM NaCl, 1 mM of EDTA, 1 mM EGTA, 1% Triton X-100, 2.5 mM sodium pyrophosphate, 1 mM beta-glycerophosphate, 1 mM Na$_3$VO$_4$, and protease inhibitor cocktail), and the concentration was measured by Pierce TM BCA Protein Assay Kit (Thermo Fisher Scientific) for normalization. The ascites from ID8 intraperitoneal OC mouse model were collected at 45 days post-injection. Peritoneal fluid washed by 1 mL PBS from mice with intraperitoneal injection of ID8 cells or PBS control was collected at 9 days post-injection. Cultured medium, ascites, and peritoneal flushing fluid were centrifuged at 1000 $\times$ g for 20 min at 4°C. The concentration of murine and human IL-18 in the supernatants was assayed by murine IL18 ELISA Kit (Wuxin Donglin Sci & Tech Development, Wuxi, Jiangsu, China) and human IL18 ELISA Kit (Wuxin Donglin Sci & Tech Development). The concentrations of murine IL-20 and IL-24 in peritoneal flushing fluid were assayed by murine IL20 ELISA Kit (Wuxin Donglin Sci & Tech Development) and murine IL24 ELISA Kit (Wuxin Donglin Sci & Tech Development). The OD (450 nm) values were measured by Infinite M200 PRO (TECAN, Männedorf, Switzerland) microplate reader. The data were analyzed by Curve Expert (Hyams Development).

## Deep RNA sequencing

Total RNA of IL20RA-reconstituted and control ID8 cells were stimulated with IL-20 for 24 hr before harvested using TRIzol reagent for RNA extraction. The deep RNA sequencing was performed and analyzed on BGI seq500 platform (BGI-Shenzhen, China). KEGG pathway analyses were conducted and analyzed based on KEGG pathway database (http://www.genome.jp/kegg/).

## ChIP-qPCR

The ChIP was performed by using ChIP-IT Express Kits according to the manufacturer's instructions (Active Motif, San Diego, CA). For immunoprecipitation, 7 µg of sheared chromatin DNA fragments were incubated with IgG or STAT3-specific antibodies. qPCR assay was used to detect the relative enrichment. The primer sequences are listed in *Supplementary file 1*.

## Statistics

Prism 8.0 software (GraphPad Software, San Diego, CA, USA) was used for statistical analysis. Statistical parameters including the definitions and exact values of n, statistical test, and statistical significance are reported in the figures and figure legends. Quantitative data were presented as means $\pm$ SEM, and the differences between the groups were analyzed using the Student's t-test. Survival curves were analyzed using the Kaplan–Meier method, and the log-rank test was used to calculate the differences between the curves. Differences are considered statistically significant at *p<0.05; **p<0.01; ***p<0.001; ns means no significance.

## Acknowledgements

The National Natural Science Foundation of China No. 81772974 (YS), the National Natural Science Foundation of China No. 81972882 (RX), and the Bilateral Inter-Governmental S&T Cooperation Project from the Ministry of Science and Technology of China 2018YFE0114300 (RX) are gratefully acknowledged.

# Additional information

## Funding

| Funder | Grant reference number | Author |
|---|---|---|
| National Natural Science Foundation of China | 81772974 | Yi Shi |
| National Natural Science Foundation of China | 81972882 | Rong Xiang |
| Chinese Ministry of Science and Technology | 2018YFE0114300 | Rong Xiang |
| Fundamental Research Funds for the Central Universities | 63211137 | Longlong Wang |

The funders had no role in study design, data collection and interpretation, or the decision to submit the work for publication.

## Author contributions

Jia Li, Conceptualization, Data curation, Formal analysis, Investigation, Writing - original draft; Xuan Qin, Resources, Data curation, Investigation, Methodology; Jie Shi, Data curation, Investigation, Methodology; Xiaoshuang Wang, Data curation, Investigation; Tong Li, Mengyao Xu, Yongjun Piao, Data curation; Xiaosu Chen, Formal analysis, Methodology; Yujia Zhao, Investigation, Methodology; Jiahao Han, Investigation; Wenwen Zhang, Pengpeng Qu, Resources; Longlong Wang, Writing - original draft; Rong Xiang, Conceptualization, Supervision, Funding acquisition, Writing - review and editing; Yi Shi, Conceptualization, Supervision, Funding acquisition, Project administration, Writing - review and editing

## Author ORCIDs

Yi Shi (iD) https://orcid.org/0000-0003-2530-410X

## Ethics

Human subjects: The patients' study was performed in accordance with the ethics committee of Nankai University and Tianjin Center Hospital of Gynecology Obstetrics (Ethic approved number: 2018KY032), and conformed to the principles embodied in the Declaration of Helsinki. All patients from Tianjin Center Hospital of Gynecology Obstetrics provided informed consent.
Animal experimentation: All of the animals were handled according to approved Institutional Animal Care and Use Committee protocols of the Nankai University. Animal experiments were approved by Institutional Animal Care and Use Committee of Nankai University (Ethic approved number: 20180014). All attempts are made to minimize the handling time during surgery and treatment so as not to unduly stress the animals. Animals are observed daily after surgery to ensure there are no unexpected complications.

## Decision letter and Author response

Decision letter https://doi.org/10.7554/eLife.66222.sa1
Author response https://doi.org/10.7554/eLife.66222.sa2

# Additional files

## Supplementary files

- Supplementary file 1. Primes for qPCR. The primers used for qRT-PCR and ChIP- qPCR analyses.

- Transparent reporting form

## Data availability

All data generated or analysed during this study are included in the manuscript and supporting files. Source data files have been provided for Figures 1-9.

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
