## [Decision Letter]

Our editorial process produces two outputs: i) public reviews designed to be posted alongside the preprint for the benefit of readers; ii) feedback on the manuscript for the authors, including requests for revisions, shown below.

Thank you for submitting your article "A systematic CRISPR screen reveals an IL-20/IL20RA-mediated immune crosstalk to prevent the ovarian cancer metastasis" for consideration by *eLife*. Your article has been reviewed by 2 peer reviewers, one of whom is a member of our Board of Reviewing Editors, and the evaluation has been overseen by Tadatsugu Taniguchi as the Senior Editor. The reviewers have opted to remain anonymous.

Essential revisions:

1. Besides IL20RA, TEX14 was highly ranked in their Crispr screen. The authors should comment if TEX14 is known to be related to IL-20RA-dependent signaling-that is, whether it fits into the same pathway.

2. They found that silencing IL20RA in SK-OV-3 cells promoted metastasis while reconstituting IL20RA in metastatic ID8 cells reduced metastatic nodules and that IL20RA was decreased in transcoelomic-spread cancer samples compared to primary sites. Thus, high IL20RA expression was beneficial. What are levels of IL20RB? Does knocking down IL20RB have similar effects to targeting IL20RA? Can the authors knock out IL20RB in human SK-OV-3 cells and/or mouse ID8 cells with IL20RA overexpression, and test them in vivo?

3. The authors used both human and mouse cell lines in a mouse model and obtained similar observations. This is potentially surprising as it suggests that the relevant pathways in human and mouse cells function even when one crosses between two species. Are human and mouse IL-18 pathways known to exhibit species specificity or to cross-react?

4. CM IL20RA-educated BMDM significantly reduced malignant ascites and nodules, indicating that IL20RA crosstalk between OC cells and macrophages limits metastasis. In macrophage depleted mice, reconstituting IL20RA in ID8 cells did not reduce metastasis, indicating that macrophages are needed for the effect. Interestingly, mesothelial cells in peritoneum produce IL20RA ligands IL20 and IL24 when OC cells are injected. Are these cytokines produced in response to other tumor cells or is the effect specific for OC cells?

5. IL20/IL20RA activates OAS1A/RNaseL mediated NLR signaling to produce IL18 mRNA and protein with activation of STAT3. There is a bigger effect on OAS1A mRNA than protein but the increase in IL-18 is robust. By ChIP-qPCR, they confirm a STAT3 binding site on the Oas1a gene but what about in the gene encoding IL-18?

6. Although not essential, more in vivo data would be valuable-e.g., using IL-20/IL-24 double KO mice and IL-18R KO mice to better confirm their findings in vivo.

7. IL20RA did not affect proliferation or migration of OC cells but resulted in increased M1-like and decreased M2-like macrophages but no change in B cells or T cells. Interestingly, IL-20 had no direct effect on M1 or M2 markers. What about IL-24?

8. In Figure 4, one might expect that the supernatants from SK-OV-3 would only affect the phenotypes of human macrophage cells, but it impacted macrophage phenotypes in vivo as shown in figure 3F. The authors should discuss/explain.

9. In Figure 4 and 5, the authors use CM from IL-20 stimulated cells. They should include controls from cells that were not treated with IL-20.

10. IL-20 and IL-24 produced by mesothelial cells acted on tumor cells. Are levels of these cytokines detectable in the ascites and normal peritoneum fluid?

11. The scale in figure 7F is misleading--it should start from 0 instead of 12. The difference is obviously small. Moreover, the difference does not appear to be biologically relevant, since in Figure 7K, the authors showed that as low as 5 ng/ml of IL-18 was sufficient to induce M1 markers.

*Reviewer #1 (Recommendations for the authors):*

1. They used the GeCKO v.2.0 CRISPR library. Besides IL20RA, TEX14 was highly ranked in metastasized OC xenografts. Is TEX14 known to be related to IL-20RA-dependent signaling?

2. They found that silencing IL20RA in SK-OV-3 cells promoted metastasis while reconstituting IL20RA in metastatic ID8 cells reduced metastatic nodules. They also found that IL20RA was decreased in transcoelomic spread cancer samples compared to primary sites. Thus, high IL20RA expression was beneficial. IL20RB was not similarly changed. Although not an essential experiment to perform, do the authors know if knocking down IL20RB would have similar effects to targeting IL20RA?

3. Mechanistically, IL20RA did not affect proliferation or migration of OC cells but resulted in increased M1-like and decreased M2-like macrophages but no change in B cells or T cells. CM from IL20RA-transduced ID8 cells stimulated expression of M1 markers and decreased M2 markers in RAW 264.7 cells and this also occurred in BM from B6 mice. In THP1 cells, CM form SK-OV-3 cells in which IL20RA is silenced induced M2 and lowered M1. Interestingly, IL-20 had no direct effect on M1 or M2 markers. Is the same true for IL-24?

4. CMIL20RA-educated BMDM significantly reduced malignant ascites and nodules, indicating that IL20RA crosstalk between OC cells and macrophages limits metastasis. In macrophage depleted mice, reconstituting IL20RA in ID8 cells did not reduce metastasis, indicating that macrophages are needed for the effect. Interestingly, mesothelial cells in peritoneum produce IL20RA ligands IL20 and IL24 when OC cells are injected. Are these cytokines also produced in response to other tumor cells or is the effect specific for OC cells?

5. IL20/IL20RA activates OAS1A/RNaseL mediated NLR signaling to produce IL18 mRNA and protein with activation of STAT3. There is a bigger effect on OAS1A mRNA than protein but the increase in IL-18 is robust. By ChIP-qPCR, they confirm a STAT3 binding site on the Oas1a gene but what about in the gene encoding IL-18? OAS1, IL-18, and pSTAT3 are lower in metastatic lesions, consistent with less IL-20 signaling. IL-18 when produced increases M1 and decreases M2 and concordant with this administering IL-18 lowers metastasis. Given the biological effect of IL-18, are there is any differences observed in the level of IL-18 receptor expression in situations of greater versus lower metastasis?

*Reviewer #2 (Recommendations for the authors):*

At the cellular and molecular level many of evidence are superficial and only supported with in vitro cell lines or cells not necessary relevant to in vivo settings, without using specific knockout mice such as IL-20/IL-24 double KO mice and IL-18R KO mice, which are essential for building the mechanistic links in vivo. In addition, the authors used both human and mouse cell lines in a mouse model and obtained same observations. This is very strange, which means that all involved pathways in human and mouse cells can perfectly function even when cross two species. This is impossible. For example, it is known human and mouse IL-18 pathways do not cross react. Mouse IL-20 also signal through mouse IL-20Ra/IL-20Rb drastically different from human IL-20Ra/IL-20Rb (Kolumam et al., Plosone 2017). Thus, the proposed mechanisms can't be true concurrently for both of human and mouse cell lines.

1) To further confirm the important role of IL-20R pathway in the transcoelomic metastatic process, the authors should also knock out IL-20Rb chain in human SK-OV-3 cell line and in mouse ID8 line with IL-20Ra overexpression, and test them in vivo.

2) The authors need to provide protein expression level of IL-20Rb.

3) In figure 4, the supernatants from SK-OV-3 could only affect the phenotypes of human macrophage cells. But clearly it impacted macrophage phenotypes in vivo as shown in figure 3F. Thus the proposed mechanisms could not be true in this case.

4) In figure 4 and 5, the authors should include controls with cells that were not treated with IL-20.

5) How the IL-20 and IL-24 produced by mesothelial cells acted on tumor cells. These cytokines should be secreted out before they function on targeting cells. The authors should measure the cytokine concentration in the ascites and normal peritoneum fluid.

6) Normally, p-Stat3 is measured minutes to few hours after cytokine stimulation. It is surprising that the authors measured pStat3 even after 24hr stimulation in Figure 7C and 7D.

7) The authors also need to provide evidence that both SK-OV-3 and ID8 cells indeed exposed to IL-20 or IL-24 in vivo in these models.

8) How tumor cells induced the expression of IL-20 and IL-24 from abdominal walls?

9) The scale in figure 7F is misleading, it should start from 0 but not 12. The difference was actually very small, although it met statistical significance. In fact, the difference was biologically irrelevant, since in Figure 7K, the authors showed that as low as 5 ng/ml of IL-18 was sufficiently induce M1 markers.

---

## [Author Response]

Essential revisions:1. Besides IL20RA, TEX14 was highly ranked in their Crispr screen. The authors should comment if TEX14 is known to be related to IL-20RA-dependent signaling-that is, whether it fits into the same pathway.

Yes, TEX14 also ranks high in our screen. We chose IL20RA instead of TEX14 for further investigation because the number of enriched sgRNAs and NES were even higher for IL20RA (number of enriched sgRNAs: 6 for IL20RA and 3 for TEX14; NES: -1.642 for IL20RA and -1.608 for TEX14). According to the literature, TEX14, with the full name as testis expressed 14, functions as an intercellular bridge forming factor in germ cells, which is required for spermatogenesis. TEX14 is an inactive kinase essential for the maintenance of stable intercellular bridges in gametes of both sexes but whose loss specifically impairs male meiosis (Greenbaum MP, *et.al*., PMID: 21669984). TEX14 acts to impede the terminal steps of abscission by competing for essential component CEP55, blocking its interaction in non-germ cells with ALIX and TSG101 (Iwamori T, *et al*., PMID: 20176808). Additionally, TEX14-interacting protein RBM44, whose localization in stabile intercellular bridges is limited to pachytene and secondary spermatocytes, may participate in processes such as RNA transport but is nonessential to the maintenance of intercellular bridge stability (Greenbaum MP, *et.al*., PMID: 21669984). There is still no evidence to show any functional relevance between TEX14 and IL20RA-dependent signaling. To further confirm, we investigated the mRNA levels of Tex14 in IL20RA-reconstituted ID8 cells and IL20RA-silenced SK-OV-3 cells. The results show that the expression of Tex14 is not affected by either the reconstitution of IL20RA or the silencing of IL20RA as shown in the following figure (Author response image 1).

**Author response image 1. sa2fig1:** The mRNA levels of Tex14 in IL20RA-reconstituted ID8 cells and IL20RA-silenced SK-OV-3 cells. The cells are treated with 2.5 ng/mL recombinant IL-20 protein before qRT-PCR analysis of Tex14 transcription.

2. They found that silencing IL20RA in SK-OV-3 cells promoted metastasis while reconstituting IL20RA in metastatic ID8 cells reduced metastatic nodules and that IL20RA was decreased in transcoelomic-spread cancer samples compared to primary sites. Thus, high IL20RA expression was beneficial. What are levels of IL20RB? Does knocking down IL20RB have similar effects to targeting IL20RA? Can the authors knock out IL20RB in human SK-OV-3 cells and/or mouse ID8 cells with IL20RA overexpression, and test them in vivo?

Thank the reviewer for the suggestion. Actually, in our previous studies, we did investigate the impact of IL20RB in the transcoelomic-spread of OC. However, these results were not added in the manuscript. As suggested, we have put these results back into the revised manuscript in the Figure 8—figure supplement 1. The results show that silencing IL20RB in IL20RA-reconsitituted ID8 cells strongly inhibits the activation of the downstream STAT3-NLR signaling (Figure 8—figure supplement 1A-B). And the conditioned medium (CM) of the IL20RB-silenced and IL20RA-reconsituted ID8 cells with IL-20 stimulation can no longer trigger the polarization of macrophages (Figure 8—figure supplement 1C). In the murine model of OC, silencing IL20RB greatly attenuates the beneficial effects of IL20RA reconstitution in inhibiting the formation of malignant ascites and the peritoneal spread of OC cells (Figure 8—figure supplement 1D-G). Taken together, these results suggest that IL-20/IL20RA signaling requires the formation of IL20RA/IL20RB heterodimer as reported (Sascha Rutz, *et al* PMID: 25421700). However, in OC patients, IL20RB expression shows no significant difference between primary and metastatic OC specimen and negatively correlates with the survival of OC patients (Figure 2—figure supplement 2), highlighting that IL20RA is the key subunit of IL20RA/IL20RB heterodimer receptor that is often dysregulated in OC cells for their peritoneal dissemination.

3. The authors used both human and mouse cell lines in a mouse model and obtained similar observations. This is potentially surprising as it suggests that the relevant pathways in human and mouse cells function even when one crosses between two species. Are human and mouse IL-18 pathways known to exhibit species specificity or to cross-react?

We thank the reviewer for this important comment. In our results, we observed the role of IL20RA in preventing the peritoneal dissemination of OC in the murine models of OC established by both human and mouse OC cells, suggesting that the cytokines involved in the process may cross-react. To further confirm, we performed the following experiments:

(1) We first investigated whether the intraperitoneal injection of SK-OV-3 cells was able to stimulate the production of IL-20 and IL-24. As shown in Figure 8—figure supplement 2A, increased expression of *Il20* and *Il24* from the abdominal wall of C57BL/6 mice upon the intraperitoneal injection of SK-OV-3 cells. In addition, murine IL-20 subfamily cytokines were also shown by Kolumam, G. et al. to be able to cross-react with human receptors. They found that murine IL-20 could induce Stat3 activity in cells expressing human IL20RA and IL20RB (Kolumam, et.al., PMID: 28125663).

(2) We further investigated whether human IL-18 was able to induce the polarization of murine macrophages. We have identified that murine IL-18 has the capacity to modulate the polarization of RAW 264.7 cells from M2-subtype to M1 subtype (Figure 7J). We used 5 ng/mL human recombinant IL-18 protein to stimulate RAW 264.7 cells for 72 h. The qRT-PCR analyses show that human IL-18 significantly stimulates the expression of M1-like markers while inhibits the expression of M2-like markers in RAW 264.7 cells (Figure 8—figure supplement 2B). In addition, to further answer the relevant question #8, we used the CM of the IL-20-stimulated control SK-OV-3 cells (CM^shCtrl^) and IL20RA-silenced SK-OV-3 cells (CM^shIL20RA^) to treat the RAW 264.7 cells for 72 h. The qRT-PCR analyses show that CM^shIL20RA^ can increase the expression of M2-like markers and inhibit the expression of M1-like markers of RAW 264.7 cells (Figure 8—figure supplement 2C). Although the efficiency of human IL-18 on RAW 264.7 cells polarization is not as significant as that of murine IL-18, it indicates that human IL-18 does react on the polarization of murine macrophages.

Taken together, these data suggest that IL-20/IL20RA signaling also works in NOD-SCID mice challenged with SK-OV-3 cells.

4. CM IL20RA-educated BMDM significantly reduced malignant ascites and nodules, indicating that IL20RA crosstalk between OC cells and macrophages limits metastasis. In macrophage depleted mice, reconstituting IL20RA in ID8 cells did not reduce metastasis, indicating that macrophages are needed for the effect. Interestingly, mesothelial cells in peritoneum produce IL20RA ligands IL20 and IL24 when OC cells are injected. Are these cytokines produced in response to other tumor cells or is the effect specific for OC cells?

Thank the reviewer for this very interesting question. The peritoneal dissemination of cancer cells also occurs in other types of tumors, such as gastric cancer, colon cancer, liver cancer and bladder carcinoma. As suggested, we injected five million of CT-26 cells (murine colon cancer cells) into the peritoneal cavity of BALB/c mice and injected five million of Hepa 1-6 cells (murine hepatocellular carcinoma cells) into the peritoneal cavity of C57BL/6 mice. The mice were sacrificed 9 days post-injection. The qRT-PCR analyses show that CT-26 cells can also induce the production of *Il20* from the peritoneal mesothelial cells (Author response image 2), while Hepa 1-6 cells show no effects on the induction of IL20RA ligands (Author response image 2). These results were further confirmed by ELISA analysis of IL-20 and IL-24 cytokines in peritoneal flushing fluid (Author response image 2,D). These data suggest that the IL-20/IL-24-mediated crosstalk between peritoneal mesothelial cells and OC cells may also occur in some other, but not all, cancer types that have the potential of peritoneal dissemination.

**Author response image 2. sa2fig2:** The expression of IL20RA ligands in the abdominal wall challenged by colon cancer cells and hepatocellular carcinoma cells. A-B. qRT-PCR analysis of IL20RA ligands from the abdominal wall in mice with the intraperitoneal injection of CT-26 cells (A) and Hepa 1-6 cells (B). C-D. ELISA measurement of IL-20 and IL-24 in peritoneal flushing fluid from mice with intraperitoneal injection of CT-26 cells (C) and Hepa 1-6 cells (D).

5. IL20/IL20RA activates OAS1A/RNaseL mediated NLR signaling to produce IL18 mRNA and protein with activation of STAT3. There is a bigger effect on OAS1A mRNA than protein but the increase in IL-18 is robust. By ChIP-qPCR, they confirm a STAT3 binding site on the Oas1a gene but what about in the gene encoding IL-18?

We thank the reviewer for the constructive suggestion. According to the reviewer’s suggestion, we further identified the STAT3 binding sites on the *Il18* gene by ChIP-qPCR. The new results have been added in the new Figure 7F,G, which shows two STAT3 binding sites on the promoter of *Il18*.

6. Although not essential, more in vivo data would be valuable-e.g., using IL-20/IL-24 double KO mice and IL-18R KO mice to better confirm their findings in vivo.

We thank the reviewer for the suggestion. We agree that the use of IL-20/IL-24 double KO mice and IL-18 KO mice may provide strong support on our findings. Currently, it is difficult for us to get such KO mice in the limited time. To further investigate the essential role of IL-18 in IL-20/IL20RA-mediated downstream signaling to suppress the peritoneal dissemination of OC in vivo, we knocked down IL-18 in IL20RA-reconsitituted ID8 cells and inject ID8^Vec^, ID8^IL20RA^ and ID8^IL20RA/shIL-18^ cells into the peritoneal cavity of C57BL/6 mice. The results show that silencing IL-18 abolishes the effects of reconstituted IL20RA in the inhibition of the malignant ascites formation and the peritoneal dissemination of OC cells. Consistently, IL-20/IL20RA-induced the polarization of peritoneal macrophages to M1-like phenotypes is also blocked by the silencing IL-18. We think these data provide additional in vivo evidence to support the essential role of IL-18 downstream IL-20/IL20RA signaling in preventing the OC dissemination. We have added the new results in the new Figure 8. The description and methods of these experiments are added in revised manuscript as well.

7. IL20RA did not affect proliferation or migration of OC cells but resulted in increased M1-like and decreased M2-like macrophages but no change in B cells or T cells. Interestingly, IL-20 had no direct effect on M1 or M2 markers. What about IL-24?

We thank the reviewer for the suggestion. We have excluded the possibility that IL-20 directly regulated the polarization of macrophages (Figure 4—figure supplement 1F). As suggested, we treated the RAW 264.7 cells with 5 ng/mL recombinant murine IL-24 protein for 72 h. The qRT-PCR analyses show that IL-24 does not affect the expression of both the M1- and M2-like markers in RAW 264.7 cells as well. We have added the new results in the new Figure 4—figure supplement 1G. The description and methods of these results are added in revised manuscript as well.

8. In Figure 4, one might expect that the supernatants from SK-OV-3 would only affect the phenotypes of human macrophage cells, but it impacted macrophage phenotypes in vivo as shown in figure 3F. The authors should discuss/explain.

Thank the reviewer for this suggestion. We used the CM of ovarian cancer cells to treat the macrophages of same species in in vitro experiments. As mentioned in the answer to the question #3, we used the SK-OV-3 CM^shCtrl^ and CM^shIL20RA^ to treat the RAW 264.7 cells for 72 h and the results are shown in Figure 8—figure supplement 2. We have performed additional experiments to show that the cytokines involved in IL20RA-mediated signaling can cross-react between mouse and human species, indicating that IL-20/IL20RA signaling also works in NOD-SCID mice injected with SK-OV-3 cells (Figure 8—figure supplement 2A-C).

9. In Figure 4 and 5, the authors use CM from IL-20 stimulated cells. They should include controls from cells that were not treated with IL-20.

We thank the reviewer for the constructive suggestion. We have added the controls in the above-mentioned figure (Figure 4B,E,H). The results show that the polarization of macrophages is not changed under the stimulation of CM from unstimulated IL20RA-reconstituted or IL20RA-silenced OC cells. IL20RA-mediated macrophage polarization needs the prior stimulation of the IL-20.

10. IL-20 and IL-24 produced by mesothelial cells acted on tumor cells. Are levels of these cytokines detectable in the ascites and normal peritoneum fluid?

We thank the reviewer for the suggestion. We have measured the amount of IL-20 and IL-24 in peritoneal flushing fluid from C57BL/6 mice with intraperitoneal injection of ID8 cells or PBS control. The results show that the amounts of IL-20 and IL-24 in peritoneal flushing fluid are significantly increased upon the injection of OC cells into the peritoneal cavity. We have added the new results in Figure 6D. The description and methods of these results are added in revised manuscript as well.

11. The scale in figure 7F is misleading--it should start from 0 instead of 12. The difference is obviously small. Moreover, the difference does not appear to be biologically relevant, since in Figure 7K, the authors showed that as low as 5 ng/ml of IL-18 was sufficient to induce M1 markers.

Thank the reviewer for this constructive comment. As mentioned in the answer to the question #6, we further investigated the essential role of IL-18 in IL-20/IL20RA-mediated downstream signaling to suppress the dissemination of ovarian cancer in vivo. We inject ID8^Vec^, ID8^IL20RA^ and ID8^IL20RA/shIL-18^ cells into the peritoneal cavity of C57BL/6 mice, respectively. And we measured the amount of IL-18 in ascites by ELISA. The new results show that IL20RA-reconstituted ID8 cells in intraperitoneal cavity cause significantly higher amount of IL-18 in ascites when compared with empty vector transfected cells. Compared with the previous data, the new results show more significant difference (nearly 10 ng/mL). In addition, the higher amount of IL-18 in ascites induced by IL20RA-reconstituted ID8 cells is significantly inhibited by silencing IL-18. We therefore combined the ELISA data in the old Figure 7F with the new ELISA data to generate the new Figure 8C.